EMBO
Molecular Medicine

# Defective glutamate and K⁺ clearance by cortical astrocytes in familial hemiplegic migraine type 2

Clizia Capuani[1,†], Marcello Melone[2,3,†], Angelita Tottene[1,†], Luca Bragina[2,3], Giovanna Crivellaro[1], Mirko Santello[4], Giorgio Casari[5], Fiorenzo Conti[2,3,6] & Daniela Pietrobon[1,7,*]

## Abstract

**Migraine is a common disabling brain disorder. A subtype of migraine with aura (familial hemiplegic migraine type 2: FHM2) is caused by loss-of-function mutations in $\alpha_2$ Na⁺,K⁺ ATPase ($\alpha_2$NKA), an isoform almost exclusively expressed in astrocytes in adult brain. Cortical spreading depression (CSD), the phenomenon that underlies migraine aura and activates migraine headache mechanisms, is facilitated in heterozygous FHM2-knockin mice with reduced expression of $\alpha_2$NKA. The mechanisms underlying an increased susceptibility to CSD in FHM2 are unknown. Here, we show reduced rates of glutamate and K⁺ clearance by cortical astrocytes during neuronal activity and reduced density of GLT-1a glutamate transporters in cortical perisynaptic astrocytic processes in heterozygous FHM2-knockin mice, demonstrating key physiological roles of $\alpha_2$NKA and supporting tight coupling with GLT-1a. Using ceftriaxone treatment of FHM2 mutants and partial inhibition of glutamate transporters in wild-type mice, we obtain evidence that defective glutamate clearance can account for most of the facilitation of CSD initiation in FHM2-knockin mice, pointing to excessive glutamatergic transmission as a key mechanism underlying the vulnerability to CSD ignition in migraine.**

**Keywords** ceftriaxone; glutamate transporter; migraine; Na⁺,K⁺ ATPase; spreading depression

**Subject Category** Neuroscience

## Introduction

Migraine is a common disabling brain disease, which manifests itself as recurrent attacks of typically throbbing and unilateral headache with certain associated features such as nausea and hypersensitivity to sensory stimuli. In a third of patients, the headache is preceded by transient focal symptoms that are most frequently visual (migraine with aura: MA). Increasing evidence supports a key role of cortical spreading depression (CSD) in migraine pathogenesis in that CSD underlies migraine aura (Lauritzen, 1994; Noseda & Burstein, 2013; Pietrobon & Moskowitz, 2013, 2014) and may trigger the headache mechanisms (Bolay et al, 2002; Ayata et al, 2006; Zhang et al, 2010, 2011; Karatas et al, 2013; Pietrobon & Moskowitz, 2014; Zhao & Levy, 2015). CSD is a slowly propagating wave of rapid nearly complete depolarization of brain cells lasting about 1 min that silences brain activity for several min (Somjen, 2001; Pietrobon & Moskowitz, 2014). The mechanisms of the primary brain dysfunction underlying the susceptibility to CSD ignition in the human brain and the onset of a migraine attack remain largely unknown and are a major open issue in the neurobiology of migraine.

Migraine has a strong multifactorial genetic component (de Vries et al, 2009; Russell & Ducros, 2011; Ferrari et al, 2015). As for many other multifactorial diseases, rare monogenic forms that phenocopy most or all the clinical features of the disorder are helpful for elucidating the molecular and cellular mechanisms of the disease. Familial hemiplegic migraine (FHM) is a rare monogenic autosomal dominant form of MA. Apart from motor features and longer duration of the aura, typical FHM attacks resemble MA attacks and both types of attacks may alternate in patients and co-occur within families (de Vries et al, 2009; Russell & Ducros, 2011; Ferrari et al, 2015).

FHM type 2 (FHM2) is caused by mutations in ATP1A2, the gene encoding the $\alpha_2$ subunit of the Na⁺,K⁺ ATPase (NKA) (De Fusco et al, 2003; Bøttger et al, 2012). The $\alpha_2$ NKA is expressed primarily in neurons during embryonic development and at the time of birth and almost exclusively in astrocytes in the adult brain (McGrail et al, 1991; Cholet et al, 2002; Ikeda et al, 2003; Moseley et al, 2003). The $\alpha_2$ NKA is thought to play an important role in K⁺ clearance during neuronal activity, but direct evidence is missing mainly because selective inhibitors allowing to distinguish the contributions

1   Department of Biomedical Sciences, University of Padova, Padova, Italy
2   Department of Experimental and Clinical Medicine, Università Politecnica delle Marche, Ancona, Italy
3   Center for Neurobiology of Aging, INRCA IRCCS, Ancona, Italy
4   Institute of Pharmacology and Toxicology, University of Zurich, Zürich, Switzerland
5   Vita-Salute San Raffaele University and San Raffaele Scientific Institute, Milano, Italy
6   Fondazione di Medicina Molecolare, Università Politecnica delle Marche, Ancona, Italy
7   CNR Institute of Neuroscience, Padova, Italy
    *Corresponding author. Tel: +39 049 827 6052; E-mail: daniela.pietrobon@unipd.it
    †These authors contributed equally to the work

of the glial $\alpha_2$ and the neuronal $\alpha_3$ NKA are still lacking (Ransom et al, 2000; D'Ambrosio et al, 2002; Larsen & MacAulay, 2014; Larsen et al, 2014). An important role of $\alpha_2$ NKA in glutamate (Glu) clearance is suggested by its colocalization with the Glu transporters (GluTs) GLT-1 and GLAST in astrocytic processes surrounding glutamatergic synapses in adult somatic sensory cortex (Cholet et al, 2002) and by the association between GluTs and $\alpha_2$ NKA in the same macromolecular complex (Rose et al, 2009). However, this association is not specific for the $\alpha_2$ NKA (Rose et al, 2009; Genda et al, 2011; Illarionava et al, 2014) and, although ouabain pharmacology indicates a preferential functional coupling of GluTs with $\alpha_2$ NKA in cultured astrocytes (Pellerin & Magistretti, 1997; but see Rose et al, 2009; Illarionava et al, 2014), the role of the $\alpha_2$ pump in clearance of synaptically released Glu during neuronal activity remains unclear, given the lack of functional data in brain slices.

FHM2 mutations cause the complete or partial loss of function of $\alpha_2$ NKA (De Fusco et al, 2003; Pietrobon, 2007; Tavraz et al, 2008, 2009; Leo et al, 2011; Bøttger et al, 2012; Schack et al, 2012; Swarts et al, 2013; Weigand et al, 2014). $\alpha_2$ NKA protein is barely detectable in the brain of homozygous knockin (KI) mice carrying the pathogenic W887R FHM2 mutation and is halved in the brain of heterozygous W887R/+ mutants (Leo et al, 2011). These FHM2-KI mice show a lower threshold for CSD induction and an increased velocity of CSD propagation, in vivo (Leo et al, 2011), similar to KI mouse models of FHM type 1 (FHM1) (van den Maagdenberg et al, 2004, 2010; Eikermann-Haerter et al, 2009) and to a mouse model of a familial advanced sleep syndrome in which all patients also suffered from MA (Brennan et al, 2013).

The mechanisms underlying an increased susceptibility to CSD in FHM2 are unknown, although impaired Glu and/or $K^+$ clearance by astrocytes have been suggested as hypothetical mechanisms (Moskowitz et al, 2004; Pietrobon, 2007).

Here, to test these hypotheses and to gain insights into the physiological role of the $\alpha_2$ NKA, we investigated the functional consequences of the W887R FHM2-causing mutation on Glu and $K^+$ clearance by cortical astrocytes during neuronal activity in acute cortical slices. Our findings indicate that the rates of Glu and $K^+$ clearance by cortical astrocytes are reduced in heterozygous W887R/+ KI mice and that the density of GLT-1a in the membrane of astrocytic processes surrounding cortical glutamatergic synapses is reduced by 50% in the FHM2 mutants. Using an in vitro model of CSD and ceftriaxone treatment in FHM2-KI mice as well as pharmacological inhibition of a fraction of GluTs in wild-type (WT) mice, we provide evidence that the defective Glu clearance can account for most of the facilitation of CSD initiation in the FHM2 mouse model.

## Results

To investigate whether the reduced membrane expression of the $\alpha_2$ NKA in heterozygous W887R/+ FHM2-KI mice causes a reduced rate of Glu clearance by cortical astrocytes during neuronal activity, we took advantage of the fact that i) Glu uptake into astrocytes by GluTs is electrogenic, and therefore can be monitored electrophysiologically, and ii) the time course of the Glu transporter current elicited in astrocytes upon extracellular neuronal stimulation in hippocampal slices (the so-called synaptically activated transporter current, STC) reflects, to some extent, the time course of Glu clearance by astrocytes and provides a relative indication of how rapidly synaptically released Glu is taken up from extracellular space (Bergles & Jahr, 1997; Diamond, 2005). The time course of the STC is also affected by the electrotonic properties of the astrocytic membrane, while axon propagation, release asynchrony, glutamate diffusion, and the kinetics of the transporters contribute insignificantly in hippocampal slices (Bergles & Jahr, 1997; Diamond, 2005); thus, the decay kinetics of the STC reflects the rate of Glu clearance filtered by the electrotonic properties of astrocytes (Diamond, 2005). Interestingly, the time course of the STC provides a measure of the rate of Glu clearance that is independent of the amount of Glu released (Diamond & Jahr, 2000; Diamond, 2005; Unichenko et al, 2012).

We measured the STC in cortical astrocytes by recording the inward current evoked in layer 1 astrocytes (held at −90 mV, close to the resting potential) by the stimulation of neuronal afferents with an extracellular electrode located in layer 1, 200 μm from the patch-clamped astrocyte, in acute slices of the somatic sensory cortex from P22-23 WT and FHM2-KI mice, in the presence of antagonists of Glu and GABA receptors (Fig 1A) (Bergles & Jahr, 1997; Bernardinelli & Chatton, 2008). The inward current evoked in astrocytes by a single pulse stimulation comprised a rapidly rising and decaying component (complete decay in few tens of ms) and a sustained component (Fig 1B, trace a). The rapidly decaying component was completely inhibited by the GluTs inhibitor TFB-TBOA (TBOA), identifying it as the STC (Fig 1B). The STC can thus be obtained by subtracting the residual current remaining in the presence of TBOA from the total inward current (trace a-b in Fig 1B) (Scimemi & Diamond, 2013). The STC can also be obtained by subtracting from the total inward current an exponential waveform (trace c in Fig 1C) that approximates the average TBOA-insensitive current (Fig 1C, top trace) obtained as described in Materials and Methods (Devaraju et al, 2013; Scimemi & Diamond, 2013). As in hippocampal slices (Diamond & Jahr, 2000; Diamond, 2005), also in barrel cortex slices the decay kinetics of the STC slowed down when the density of GluTs was reduced by subsaturating concentrations of DL-TBOA (Appendix Fig S1A), as expected for a reduced rate of Glu clearance and a longer lifetime of synaptically released Glu in the extracellular space. Moreover, changing the intensity of extracellular stimulation (and hence the number of stimulated fibers and the amount of Glu released) changed the STC amplitude without affecting the STC decay kinetics (Appendix Fig S1B). Thus, to obtain a measure of the rate of clearance of synaptically released Glu by cortical astrocytes, which is not affected by the amount of Glu released by extracellular stimulation, we fitted with an exponential function the decay of the STC (Fig 1).

The STC isolated pharmacologically in cortical astrocytes of WT mice decayed with an exponential time course with time constant $\tau = 6.80 \pm 0.19$ ms (Fig 1B). The STC isolated in a larger number of cells using the exponentially rising function that approximates the average TBOA-insensitive current (Fig 1C) decayed with a similar time course ($6.46 \pm 0.13$ ms; unpaired t-test, $P = 0.15$). The decay kinetics of the STC elicited in cortical astrocytes of heterozygous W887R/+ FHM2-KI mice were slower compared to those in WT astrocytes, as shown by the significantly larger time constant of decay of both the STC isolated pharmacologically ($\tau = 7.97 \pm 0.31$ ms) and the STC isolated nonpharmacologically in a larger

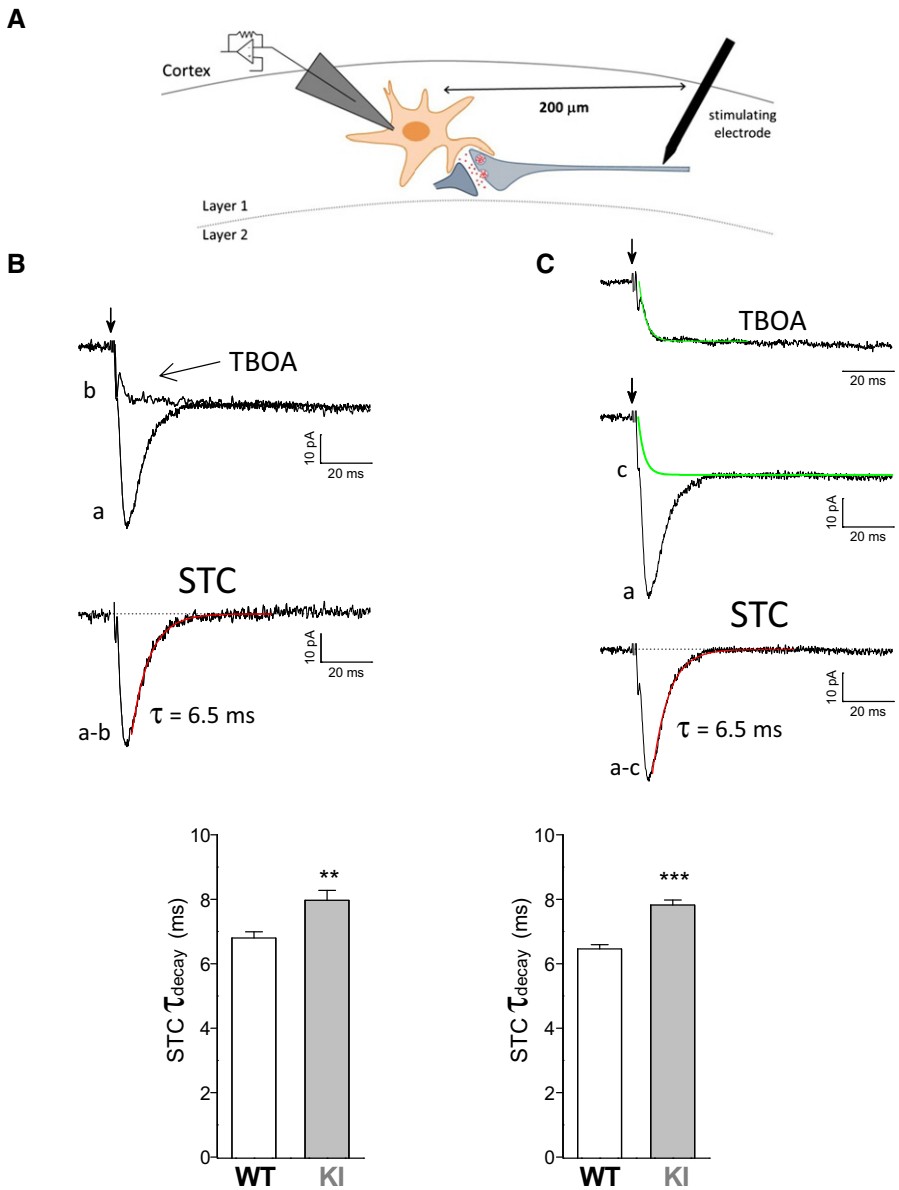

**Figure 1.  The rate of glutamate clearance by cortical astrocytes, as deduced from the decay kinetics of the synaptically activated glutamate transporter current (STC) elicited by single pulse stimulation, is slower in W887R/+ FHM2-knockin (KI) relative to wild-type (WT) mice.**

A  Scheme of the STC recording paradigm. The currents elicited by extracellular stimulation in layer 1 were measured in a voltage-clamped layer 1 astrocyte located at 200 μm from the stimulating electrode in an acute slice of mouse barrel cortex.

B  Time constants of decay, $\tau_{decay}$, of the STC isolated pharmacologically in WT and FHM2-KI mice. Top, superimposed representative traces of the inward current evoked in an astrocyte (held at −90 mV) by a single pulse stimulation (indicated by the arrow) in a WT slice, before (trace a) and after (trace b) application of a saturating concentration of the GluT inhibitor TFB-TBOA (TBOA). The STC was obtained by subtracting the residual current remaining in the presence of TBOA from the total inward current (trace a–b); the decay of the STC was best fitted by a single exponential function with $\tau_{decay}$ = 6.53 ms (in red). The bar plot shows the average values of $\tau_{decay}$ of the STC isolated pharmacologically in cortical slices (*n* = 13) from P22-23 WT (*N* = 7) and KI mice (*n* = 9; *N* = 3). STC $\tau_{decay}$ is 17% higher in KI compared to WT astrocytes (unpaired *t*-test: **P = 0.003). Hereafter, *n* indicates the number of slices and *N* indicates the number of mice. Data are mean ± SEM.

C  $\tau_{decay}$ of the STC isolated using an exponential waveform approximating the average TBOA-insensitive current in WT and KI mice. Top trace, average normalized TBOA-insensitive current obtained by pooling the normalized TBOA-insensitive currents recorded in 16 WT and KI cells. This current was best fitted by an exponentially rising function [1-exp (−*t*/$\tau_{rise}$)] with $\tau_{rise}$ = 2.35 ms (in green). The STC was obtained by subtracting from the total current elicited in the astrocyte [a: same representative trace as in (B)] the exponential function A[1-exp (−*t*/$\tau_{rise}$)] with $\tau_{rise}$ = 2.35 ms and A equal to the maximal current measured in the astrocyte at about 60 ms after stimulation (trace c); the decay of the STC (trace a–c) was best fitted by a single exponential function with $\tau_{decay}$ = 6.49 ms (in red). The bar plot shows the average values of $\tau_{decay}$ of the STC isolated as shown in the top panel in cortical slices from P22-23 WT (*n* = 28; *N* = 11) and KI mice (*n* = 27; *N* = 9). STC $\tau_{decay}$ is 21% higher in KI compared to WT astrocytes (unpaired *t*-test: ***P < 0.0001). Data are mean ± SEM.

Source data are available online for this figure.

number of cells ($\tau = 7.82 \pm 0.16$ ms) (Fig 1). The slower decay of the STC in FHM2-KI cortical slices indicates that the rate of clearance of synaptically released Glu by cortical astrocytes is slower in FHM2-KI compared to WT mice. The alternative interpretation that the slower decay of the STC in FHM2-KI mice reflects a larger electrotonic filtering in KI compared to WT astrocytes appears quite unlikely, given the similar passive membrane properties measured in KI and WT cortical astrocytes (see Materials and Methods) and the unlikelihood that the astrocyte electrotonic properties are affected by a 50% reduction in $\alpha_2$ NKA expression. Moreover, in the unlikely case that, in contrast with the findings in hippocampal slices (Diamond, 2005), other factors, such as release asynchrony or axonal propagation, contribute significantly to the time course of the STC in cortical slices, the alternative interpretation that a larger contribution of these factors in FHM2-KI mice accounts for the slower decay of the STC in KI slices seems also quite unlikely, given that the FHM2 mutation affects a specific astrocytic NKA that is not expressed in cortical axons (Cholet et al, 2002).

We next investigated whether the slowing down of Glu clearance in FHM2-KI mice was larger after repetitive stimulation with trains of 10 pulses at 50 or 100 Hz, as, for example, might be expected if the binding capacity of the GluTs in layer 1 astrocytes is overwhelmed during the train. In this case, the decay kinetics of the STC elicited by the 10th pulse in the train should be slower than those of the STC elicited by a single pulse stimulation (Diamond & Jahr, 2000). Indeed, in experiments in which single stimuli were alternated with trains of 9 and 10 pulses at 50 or 100 Hz, the time constants of decay of the STCs elicited by the 10th pulse of the high-frequency trains (isolated as described in Materials and Methods: Fig 2A) were significantly larger than those of the corresponding STC elicited by a single pulse. For example, the time constant of the STC elicited in WT astrocytes by the 10th pulse of a 50-Hz train, $\tau_{10\ (50\ Hz)}$, was $7.56 \pm 0.16$ ms ($n = 23$), while that elicited by a single pulse in the same cells was $\tau_1 = 6.45 \pm 0.15$ ms ($n = 23$; paired $t$-test, $P < 0.0001$) and the average ratio $\tau_{10}/\tau_1$ (50 Hz) was $1.18 \pm 0.02$ ($n = 23$). The slowing of the STC after the train was larger in FHM2-KI compared to WT mice (in KI: $\tau_{10}/\tau_1$ (50 Hz) = $1.26 \pm 0.03$, $n = 21$; unpaired $t$-test, $P = 0.021$). The relative slowing and the difference between WT and KI mice were more pronounced after a 100-Hz train ($\tau_{10}/\tau_1$ (100 Hz) = $1.25 \pm 0.03$, $n = 18$, in WT versus $\tau_{10}/\tau_1$ (100 Hz) = $1.40 \pm 0.05$, $n = 14$, in KI, $P = 0.009$). As a consequence, the slowing down of Glu clearance in FHM2-KI compared to WT mice was quantitatively larger after repetitive stimulation than after single pulse stimulation and increased with increasing stimulation frequency (Fig 2B). The time constants of decay of the STCs elicited by the 10th pulse of 50- and 100-Hz trains in FHM2-KI mice ($\tau_{10(50\ Hz)} = 9.82 \pm 0.24$; $\tau_{10(100\ Hz)} = 11.08 \pm 0.41$ ms) were 30 and 37% larger than those in WT mice, respectively ($\tau_{10(50\ Hz)} = 7.56 \pm 0.16$ ms; $\tau_{10(100\ Hz)} = 8.09 \pm 0.23$ ms) (Fig 2B). For comparison, the time constant of decay of the STC elicited by a single pulse was 21% larger in FHM2-KI compared to WT mice (Fig 1).

Given the evidence of association in the same protein complex of the glial GluTs (GLT-1 and GLAST) and the $\alpha_2$ NKA pump (Rose et al, 2009; Illarionava et al, 2014) and of colocalization of the glial GluTs with $\alpha_2$ NKA in astrocytic processes surrounding neocortical glutamatergic synapses (Cholet et al, 2002), we investigated whether the reduced expression of $\alpha_2$ NKA leads to a reduced

density of GLT-1 in the membrane of astrocytic processes surrounding cortical excitatory synapses (perisynaptic astrocytic processes: PAPs). GLT-1 is the quantitatively dominant GluT in the brain and mediates the majority of Glu clearance in the adult murine neocortex (Haugeto et al, 1996; Rothstein et al, 1996; Tanaka et al, 1997; Danbolt, 2001; Campbell et al, 2014).

We obtained a first indication that GLT-1 expression in the vicinity of cortical glutamatergic synapses was decreased in FHM2-KI mice from double-labeling immunofluorescence of GLT-1a and the vesicular glutamate transporter VGLUT1 in neocortical sections. GLT-1a is the predominant brain GLT-1 isoform (Chen et al, 2004; Berger et al, 2005; Holmseth et al, 2009), and VGLUT1 is expressed in the large majority of cortical excitatory terminals (Kaneko et al, 2002). Quantitative analysis of GLT-1a immunoreactivity (ir) showed that the mean size of the GLT-1a-positive (GLT-1a[+]) ir puncta (green) that overlapped with VGLUT1[+] ir puncta (red) was reduced by 18% in FHM2-KI mice ($0.47 \pm 0.03$ $\mu m^2$) compared to WT ($0.57 \pm 0.03$ $\mu m^2$) (Fig 3). The percentages of GLT-1a[+] puncta overlapping with VGLUT1[+] puncta were comparable ($49 \pm 3\%$ in KI; $46 \pm 4\%$ in WT). As a change in size of ir puncta has been considered strongly suggestive of a change in protein expression (Bozdagi et al, 2000; Bragina et al, 2006; Omrani et al, 2009), the reduced size of the GLT-1a[+] puncta overlapping with VGLUT1[+] puncta is consistent with a reduced GLT-1a expression in the vicinity of cortical glutamatergic synapses in W887R/+ KI mice.

The majority of overlapping GLT-1a/VGLUT1 puncta colocalized with $\alpha_2$ NKA ($81 \pm 2\%$), as revealed by triple-labeling immunofluorescence of GLT-1a, VGLUT1, and $\alpha_2$ NKA in WT cortical sections (Fig EV1). This suggests a strict colocalization of GLT-1a and $\alpha_2$ NKA in the astrocytic processes close to glutamatergic synapses, given that in the adult brain $\alpha_2$ NKA is expressed almost exclusively in astrocytes and is not present in cortical axon terminals (McGrail et al, 1991; Cholet et al, 2002).

We then used post-embedding immunogold electron microscopy (EM) in cerebral cortex sections to study the density of GLT-1a gold particles associated with the membrane of PAPs. In line with previous studies (e.g. Melone et al, 2009; Omrani et al, 2009), GLT-1a particles were at the membrane and in the cytoplasm of both PAPs (Fig 4A) and axon terminals (AxT) forming asymmetric synaptic contacts (Fig 4B); the density of gold particles was significantly higher on the plasma membrane (i.e. within 15 nm of the extracellular side of the membrane) than in the cytoplasm (Table 1). The total and plasma membrane densities of GLT-1a gold particles in PAPs were reduced by 41 and 48%, respectively, in FHM2-KI compared to WT mice (Fig 4A, Table 1). The reduction in GLT-1a density in the membrane of PAPs is quantitatively similar to the reduction in $\alpha_2$ NKA protein level in cortical crude synaptic membranes from W887R/+ KI mice revealed by Western blotting (about 50%: Leo et al, 2011). Interestingly, in contrast with the decreased density of GLT-1a in PAPs, both the total and the plasma membrane densities of GLT-1a were unaltered in FHM2-KI AxTs (Fig 4B, Table 1). The similar density of GLT-1a in FHM2 and WT AxTs correlates with, and likely reflects, the absence of $\alpha_2$ NKA in cortical AxTs (Cholet et al, 2002).

The EM data, together with the strict colocalization of $\alpha_2$ NKA and GLT-1a in the vicinity of glutamatergic synapses (Fig EV1), suggest a necessary tight coupling between $\alpha_2$ NKA and GLT-1a in PAPs and point to the reduced expression of GluTs in PAPs as the

   

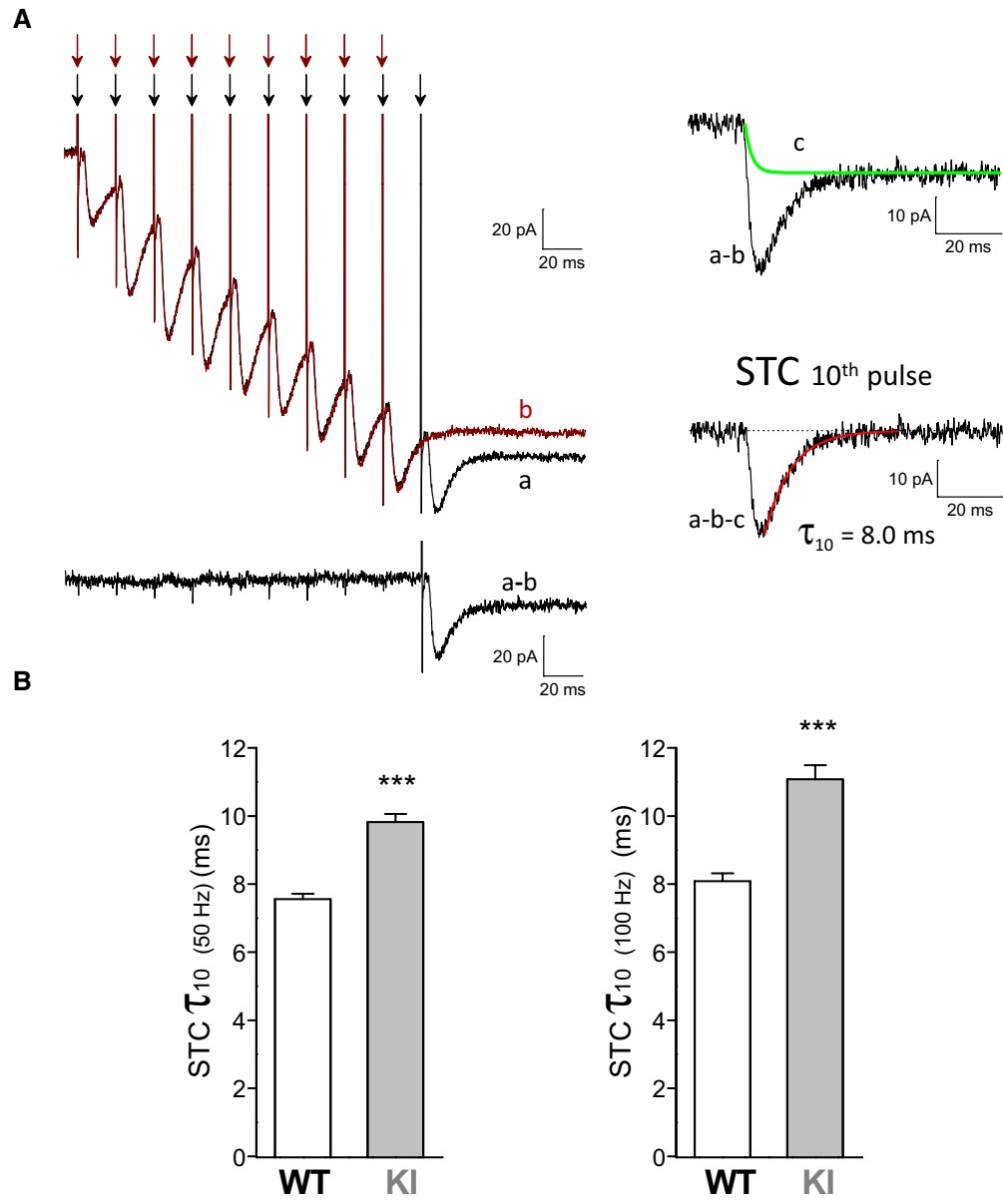

**Figure 2.  The slowing down of glutamate clearance in FHM2-KI relative to WT mice is larger after a train of action potentials at high frequency than after a single action potential.**

A   Isolation of the STC elicited by the last pulse of a high-frequency train of action potentials. Left: superimposed representative traces of the inward current evoked in an astrocyte (held at −90 mV) by extracellular stimulation with a train of 10 pulses (trace a: black) and a train of nine pulses (trace b: brown) at 50 Hz in a WT cortical slice. The inward current elicited by the 10th pulse was obtained by subtracting the current elicited by nine pulses from that elicited by 10 pulses (trace a-b). Right: The STC elicited by the 10th pulse (trace a-b-c) was obtained by subtracting the exponential function that simulates the TBOA-insensitive current elicited by a single pulse (trace c, obtained as in Fig 1C) to the 10-9 pulses difference current (trace a-b). The decay of the STC elicited by the 10th pulse was best fitted by a single exponential function with $\tau_{decay}$ = 8.04 ms (in red).

B   $\tau_{decay}$ of the STC elicited by the 10th pulse of 50-Hz ($\tau_{10\ (50\ Hz)}$, left panel) and 100-Hz ($\tau_{10\ (100\ Hz)}$, right panel) trains in layer 1 astrocytes in acute cortical slices from P22-23 WT ($n$ = 23; $N$ = 10 for 50 Hz; $n$ = 18; $N$ = 8 for 100 Hz) and KI ($n$ = 21; $N$ = 9 for 50 Hz and $n$ = 14; $N$ = 7 for 100 Hz) mice. The STC elicited by the 10th pulse was obtained as described in (A). STC $\tau_{10\ (50\ Hz)}$ and $\tau_{10\ (100\ Hz)}$ are 30% and 37% higher in KI compared to WT mice, respectively (unpaired *t*-test: ***$P$ < 0.0001 in both cases). Data are mean ± SEM.

Source data are available online for this figure.

main mechanism underlying the reduced rate of Glu clearance by astrocytes in FHM2-KI mice.

To investigate whether the reduced expression of the $\alpha_2$ NKA in W887R/+ KI mice causes a reduced rate of K$^+$ clearance by astrocytes during neuronal activity, we took advantage of the fact that (i) as a consequence of the very high expression of K$^+$ channels that are open at negative resting voltages, the astrocyte passive conductance is essentially K$^+$ selective and the astrocyte membrane

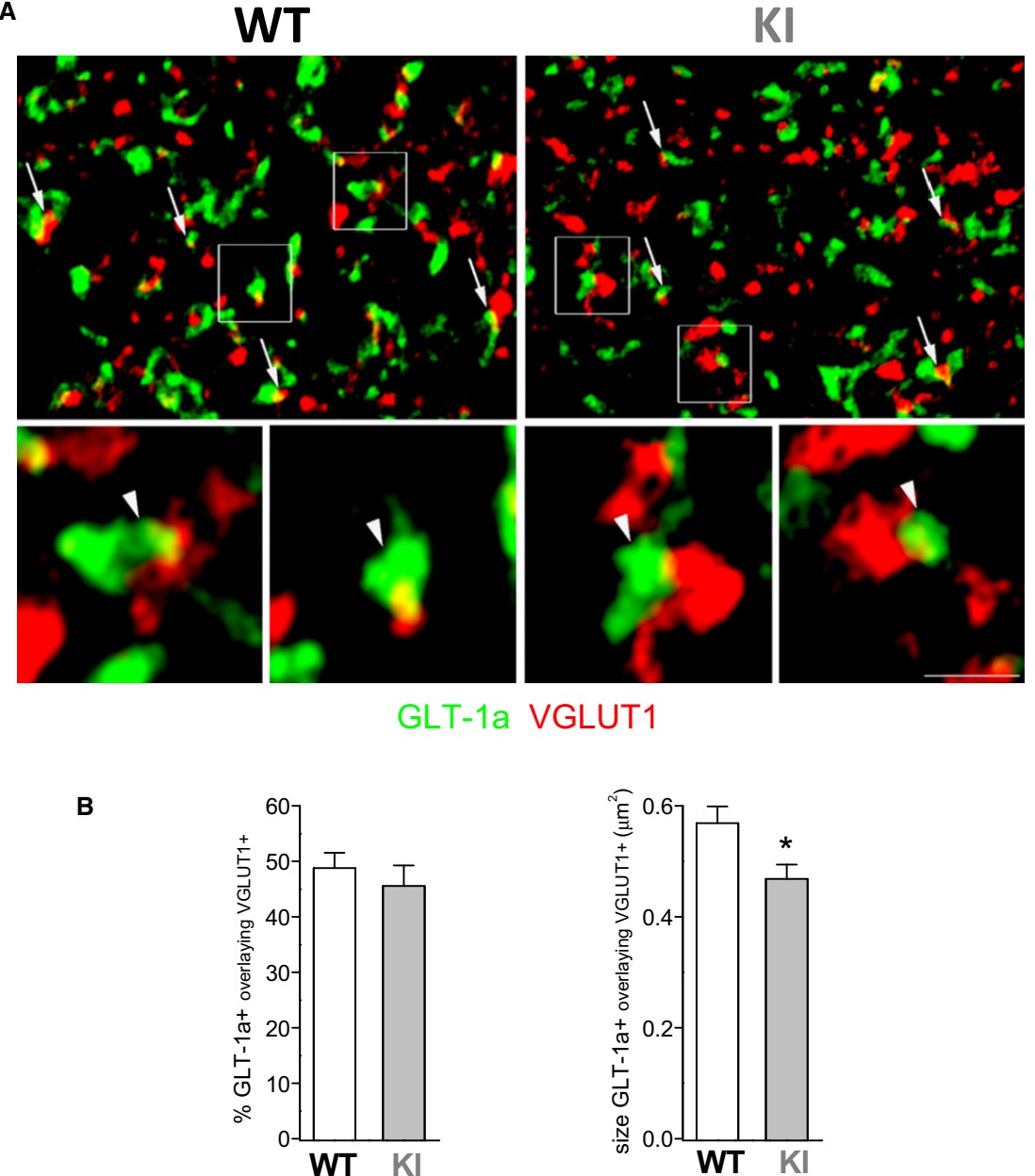

GLT-1a  VGLUT1

**Figure 3.  The size of GLT-1a-immunoreactive (ir) puncta overlapping with VGLUT1 ir puncta is reduced in the cortex of W887R/+ FHM2-KI mice, suggesting a reduced expression of the glutamate transporter GLT-1a in the vicinity of cortical glutamatergic synapses.**

A    Simultaneous visualization of GLT-1a[+] (green) and VGLUT1[+] puncta (red) in first somatic sensory cortex (SI) of a WT and a KI mouse (P35). Arrows point to some GLT-1a[+] puncta overlaying with VGLUT1[+] puncta (i.e. GLT-1a/VGLUT1-related puncta); framed regions (enlarged below) are examples of GLT-1a/VGLUT1-related puncta (arrowheads). All microscopic fields are from layers II/III. Scale bar: 3.5 μm for left and right upper panels and 1 μm for enlarged framed areas.

B    Percentage and size of GLT-1a[+] puncta overlaying with VGLUT1 in P35 WT and KI mice. Left, percentage of GLT-1a[+] puncta overlaying with VGLUT1 is comparable in WT and KI mice (data were obtained from 14 and 18 fields of 20 × 20 μm from 2 WT and 2 KI mice (four sections/animal), respectively) (unpaired *t*-test: *P* = 0.52). Right, size of GLT-1a/VGLUT1-related puncta is 18% reduced in KI mice (169 GLT-1a[+] puncta analyzed from 2 mice) compared to WT (174 GLT-1a[+] puncta analyzed from 2 mice) (Mann–Whitney *U*-test: **P* = 0.016). Data are mean ± SEM.

Source data are available online for this figure.

potential behaves as a good K[+] electrode, rendering astrocytes useful [K[+]]$_e$ biosensors (Meeks & Mennerick, 2007; Zhou *et al*, 2009; Hwang *et al*, 2014); (ii) the slowly decaying inward current elicited in astrocytes upon extracellular neuronal stimulation in acute brain slices is largely a K[+] current that reflects [K[+]]$_e$ accumulation due to neuronal K[+] efflux and the accompanying change in driving force through glial K[+] channels (mainly inward rectifier Kir channels judging from the sensitivity of the slow current to

low $Ba^{2+}$ concentrations and to Kir4.1 knockout) (De Saint & Westbrook, 2005; Djukic *et al*, 2007; Meeks & Mennerick, 2007; Bernardinelli & Chatton, 2008; Shih *et al*, 2013; Sibille *et al*, 2014). The time constant of decay of this slow current (hereafter called $I_K$) provides a measure of the rate of $K^+$ clearance, which has been shown to be equivalent to that obtained with $[K^+]_e$-sensitive microelectrodes (Meeks & Mennerick, 2007).

Therefore, to investigate the effect of the FHM2 mutation on $K^+$ clearance, we recorded the current evoked in layer 1 astrocytes by extracellular stimulation in layer 1, a protocol similar to that used for the STC. However, the recordings were performed in the absence of synaptic receptor blockers because postsynaptic Glu receptors, in

particular the NMDA receptors (NMDARs), represent a major source of $K^+$ efflux during neuronal activity (Poolos *et al*, 1987; De Saint & Westbrook, 2005; Shih *et al*, 2013; Sibille *et al*, 2014). The $I_K$ current elicited by repetitive stimulation (10 pulses at 50 Hz) in WT astrocytes decayed with a time constant of $2.36 \pm 0.10$ s, more than two orders of magnitude more slowly than the STC (Fig 5). The decay kinetics of $I_K$ elicited by the same stimulation in FHM2-KI cortical slices were slower, as shown by the 22% larger time constant ($2.87 \pm 0.10$ s) (Fig 5). This indicates that the reduced membrane expression of the $\alpha_2$ NKA in W887R/+ FHM2-KI mice causes a reduced rate of $K^+$ clearance by cortical astrocytes following neuronal activity. The measurement of the time constant of

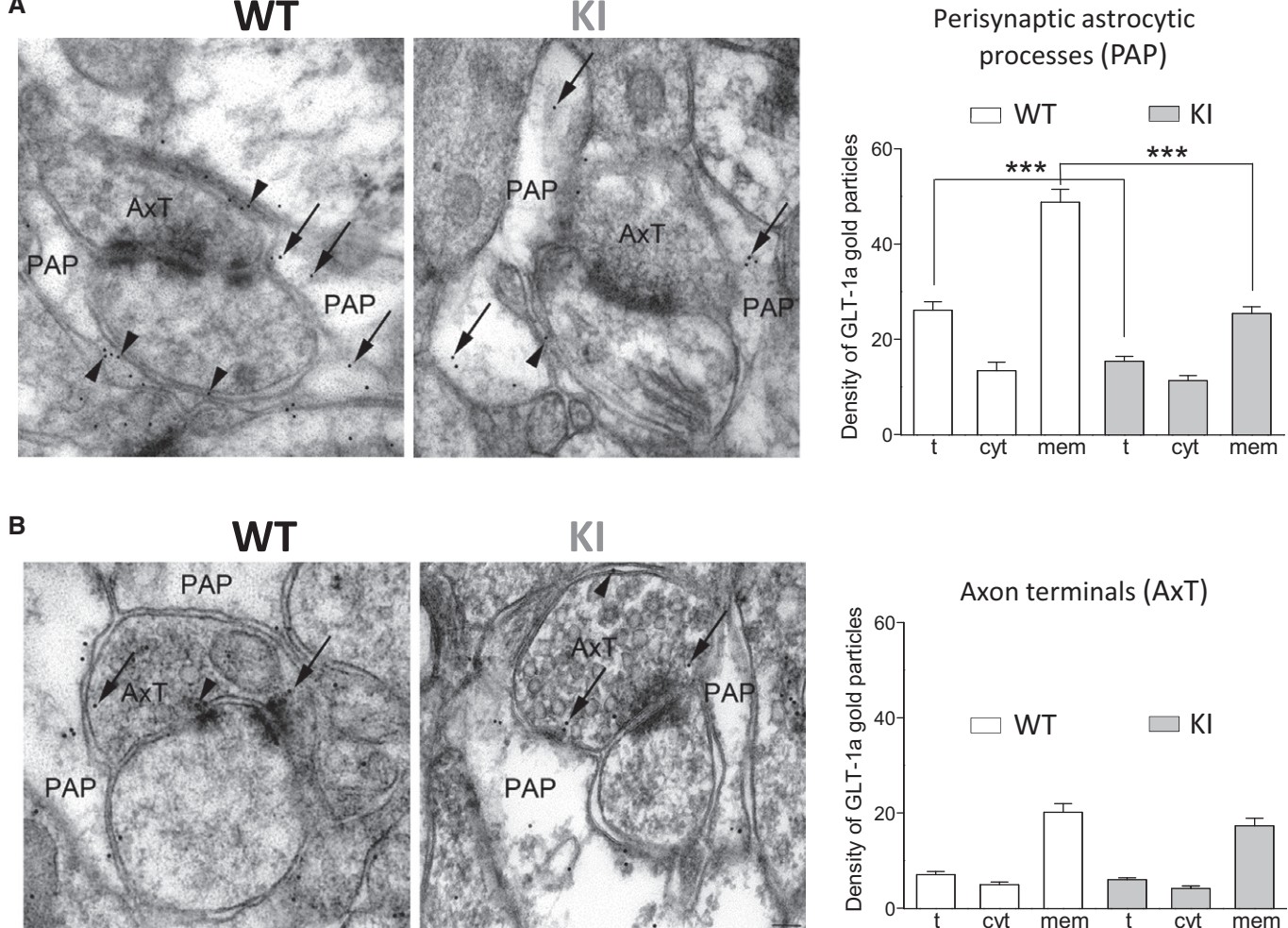

**Figure 4. The density of GLT-1a in the membrane of cortical perisynaptic astrocytic processes is reduced in W887R/+ FHM2-KI mice, while the density of GLT-1a in axon terminals is unaltered.**

A   Distribution of GLT-1a gold particles in astrocytic processes contacting asymmetric synapses (perisynaptic astrocytic processes: PAP) in SI of 5 WT and 5 KI (two male, three female) mice (P34–35). In cortical PAP of KI mice, the density (particles/$\mu m^2$) of total and membrane-associated gold particles (arrowheads) are reduced compared to WT mice, whereas the density of cytoplasmic gold particles (arrows) is comparable to that in WT (see Table 1 for numerical values and statistics).

B   Distribution of GLT-1a gold particles in excitatory axon terminals forming asymmetric synaptic contact (AxT) in SI of P34-35 WT and KI mice. In cortical AxT of KI and WT mice, GLT-1a density is comparable (arrowheads: membrane-associated gold particles; arrows: cytoplasmic gold particles) (see Table 1 for statistics).

Data information: All microscopic fields in (A) and (B) are from layers II/III. t, total density; cyt, cytoplasmic density; mem, membrane density. Scale bar: 100 nm. Data are mean $\pm$ SEM.

Source data are available online for this figure.

**Table 1.   Density of GLT-1a gold particles in perisynaptic astrocytic processes and axon terminals of asymmetric synapses of WT and W887R/+ FHM2-KI mice.**

Density values are mean $\pm$ SEM; $n$ = the number of profiles. Comparison of densities between WT ($N$ = 5) and KI ($N$ = 5) and comparison of densities of perisynaptic astrocytic processes, axon terminals, and background was made using Mann–Whitney $U$-test.

| Localization | WT (particles/$\mu m^2$) | FHM2 KI (particles/$\mu m^2$) | WT versus FHM2 KI |
|---|---|---|---|
| Nucleus (background) | $0.59 \pm 0.02$ ($n$ = 30) | $0.68 \pm 0.03$ ($n$ = 30) | |
| Astrocytic processes° | $26.14 \pm 1.80$ ($n$ = 481) | $15.39 \pm 1.04$ ($n$ = 524) | $P < 0.0001$ |
| Plasma membrane | $48.79 \pm 2.66$ | $25.40 \pm 1.38$ | $P < 0.0001$ |
| Cytoplasm | $13.40 \pm 1.76$ | $11.26 \pm 1.06$ | $P = 0.058$ |
| Axon terminals° | $7.05 \pm 0.65$ ($n$ = 148) | $6.00 \pm 0.43$ ($n$ = 121) | $P = 0.44$ |
| Plasma membrane | $20.15 \pm 1.77$ | $17.34 \pm 1.57$ | $P = 0.47$ |
| Cytoplasm | $4.94 \pm 0.60$ | $4.14 \pm 0.46$ | $P = 0.45$ |

°density of gold particles was significantly higher than background in PAPs and AxTs ($P < 0.0001$ for both WT and KI). The mean areas of PAP and AxT of all sampled profiles used for immunogold analysis were comparable in WT and KI (PAP: $0.33 \pm 0.09$ versus $0.38 \pm 0.05$ $\mu m^2$; Mann–Whitney $U$-test, $P = 0.53$; AxT: $0.71 \pm 0.17$ versus $0.56 \pm 0.10$ $\mu m^2$; Mann–Whitney $U$-test, $P = 0.55$).

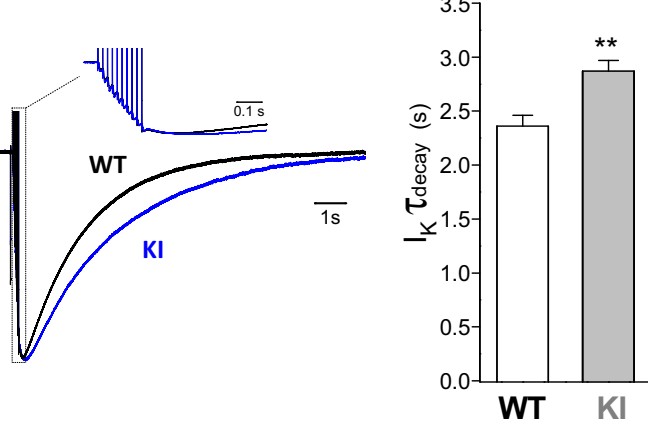

**Figure 5.   The rate of $K^+$ clearance by cortical astrocytes after a train of action potentials is reduced in W887R/+ FHM2-KI compared to WT mice.**
Left, superimposed representative traces of the normalized inward current evoked in a WT (black trace) and a KI cortical astrocyte (blue trace) by extracellular stimulation with a train of 10 pulses at 50 Hz in cortical slices. The inset shows in an expanded time scale the portion of the traces indicated by the dotted line. The slowly decaying current ($\tau_{decay}$ = 2.00 s and 3.04 s for the WT and KI trace, respectively) is largely a $[K^+]_e$-dependent $K^+$ current ($I_K$) whose decay kinetics provide a measure of the rate of $K^+$ clearance by astrocytes (see text). Right, time constant of decay of $I_K$ elicited by trains of 10 pulses at 50 Hz in cortical astrocytes from P22-23 WT ($n$ = 21; $N$ = 12) and KI mice ($n$ = 20; $N$ = 8). $I_K$ $\tau_{decay}$ is 22% higher in KI compared to WT astrocytes (unpaired $t$-test: **$P = 0.001$). Data are mean $\pm$ SEM.

Source data are available online for this figure.

decay of $I_K$ following stimulation with 10 pulses at 50 Hz in the presence of Glu receptor blockers in WT ($I_K$ $\tau_{decay}$ = $2.38 \pm 0.09$ s, $n$ = 18) and FHM2-KI astrocytes ($I_K$ $\tau_{decay}$ = $2.95 \pm 0.08$ s, $n$ = 16) confirms this conclusion. In both genotypes, the time constants of decay of $I_K$ are similar in the absence and presence of Glu receptors blockers, and the slowing of $K^+$ clearance in FHM2-KI cortical slices is also similar in the two conditions.

We next investigated whether the slower rate of Glu clearance during neuronal activity contributes to the facilitation of CSD in W887R/+ FHM2-KI mice (Leo $et$ $al$, 2011). First, we verified that in

the conditions in which we recorded the STC (P22 – 23 mice, $T$ = 30°C), CSD induced in acute cortical slices by brief pulses of high $K^+$ (as in Tottene $et$ $al$, 2009) was facilitated in FHM2-KI mice, as shown $in$ $vivo$ in adult mice (Leo $et$ $al$, 2011). Brief pressure-ejection KCl pulses of increasing duration were applied (at 5-min intervals) onto the slice surface (layer 2/3) until a CSD was elicited, as revealed by the associated changes in intrinsic optic signal (IOS). The duration of the first pulse eliciting a CSD was taken as CSD threshold and the rate of horizontal spread of the change in IOS as CSD velocity (Fig 6A). The threshold for CSD induction was 28% lower in FHM2-KI mice relative to WT ($170 \pm 4$ versus $236 \pm 8$ ms) and the velocity of CSD propagation was 21% higher ($3.94 \pm 0.04$, versus $3.26 \pm 0.05$ mm/min) (Fig 6B). Thus, experimental CSD in acute cortical slices from FHM2-KI mice was facilitated in the conditions in which a slowing of the rate of Glu clearance compared to WT was shown. The smaller extent of facilitation compared to $in$ $vivo$ situation (where the CSD threshold was decreased about 50% and the velocity increased about 40%) is at least in part due to the younger age of the animals and the lower temperature. In fact, the facilitation of CSD was larger in cortical slices from older (P34 - 35) mice: the threshold for CSD induction ($T$ = 30°C) was 39% lower and the velocity of CSD propagation 26% higher in FHM2-KI relative to WT mice (Appendix Fig S2A). In contrast with the very small number of spontaneous CSDs in both WT and KI slices from P22 – 23 mice (2 out of 26 and 1 out of 28, respectively), it was necessary to increase the perfusion rate of P34 – 35 cortical slices (to 13 ml/min) to prevent the frequent ignition of spontaneous CSDs in KI slices; in the latter, the frequency of spontaneous CSDs (5 out of 31) remained larger than in WT (1 out of 21) even at this high flow rate. On the other hand, the facilitation of CSD was smaller in cortical slices from P34 – 35 mice at lower temperature: the threshold for CSD induction (room T) was 22% lower and the velocity of CSD propagation 20% higher in FHM2-KI relative to WT mice (Appendix Fig S2B); no spontaneous CSDs were observed.

As a first approach to study whether there is a causative relationship between reduced rate of Glu clearance by astrocytes and CSD facilitation, we investigated whether the facilitation of CSD in the FHM2-KI mice could be (at least partially) rescued by systemic treatment with ceftriaxone (Cef), a drug that increases the membrane

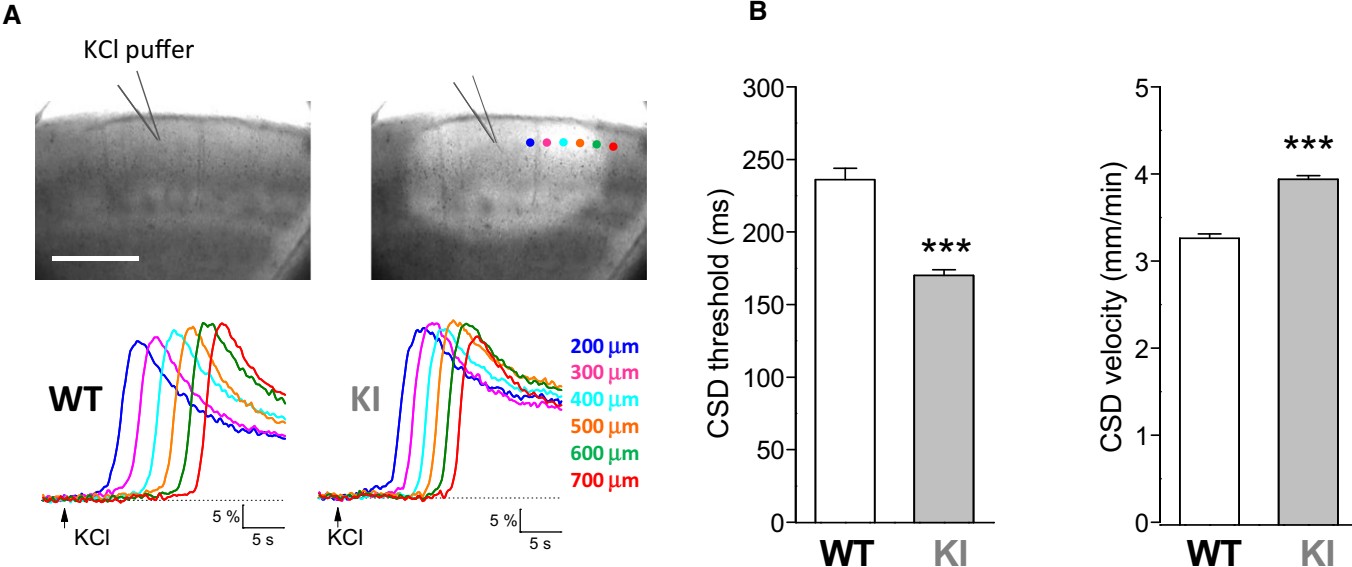

**Figure 6.  Facilitation of CSD induction and propagation in acute cortical slices of W887R/+ FHM2-KI mice.**

A   Images of a cortical slice before ($t = 0$) and after ($t = 15.6$ s) pressure ejection of a high KCl pulse that elicits a CSD (top panels), showing that the propagating CSD is associated with a change in light transmittance. Scale bar: 500 μm. The traces below show representative changes in intrinsic optical signal (IOS) relative to background measured in a WT and a KI cortical slice during CSD propagation at increasing times and distances from the KCl puff, as indicated in color code in the right image on the top. The velocity of CSD propagation, obtained from the rate of horizontal spread of the change in IOS, in these two representative WT and KI slices is 3.11 and 4.14 mm/min, respectively.

B   Stimulation threshold for CSD induction (CSD threshold) and rate of CSD propagation (CSD velocity) in WT ($n = 24$; $N = 3$) and KI ($n = 27$; $N = 8$) cortical slices from P22-23 mice. CSD threshold is expressed as duration of the first KCl pulse eliciting a CSD. CSD threshold is 28% lower (Mann–Whitney $U$-test, ***$P < 0.0001$) and CSD velocity 21% higher (unpaired $t$-test: ***$P < 0.0001$) in KI compared to WT mice. Data are mean ± SEM.

Source data are available online for this figure.

expression of GLT-1 in neocortex (Bellesi *et al*, 2009). Indeed, Western blotting of cortical crude synaptic membranes and double-labeling immunofluorescence of GLT-1a and VGLUT1 in cortical sections from WT mice that had been injected for 8 days with either Cef (200 mg/kg) or saline showed a 63% increase in GLT-1a protein (Fig 7A) and a 58% increase in the mean size of the GLT-1a$^+$ puncta overlapping with VGLUT1 in Cef-treated compared to saline-treated control animals ($0.84 \pm 0.03$ μm$^2$ versus $0.53 \pm 0.03$ μm$^2$; Fig 7B). Threshold of CSD induction and velocity of CSD propagation were measured in slices from P30-33 FHM2-KI mice that had been injected for 7–8 days with either Cef or saline. At this age, the frequency of spontaneous CSDs was relatively small (in 7 out of 47 slices from saline-injected KI mice) even using 6 ml/min perfusion rate. Moreover, at this age, $\alpha_2$ NKA and GLT-1 are expressed at very close to adult levels (Orlowski and Lingrel, 1988; Furuta *et al*, 1997; Ullensvang *et al*, 1997). The threshold for CSD induction was slightly (12%), but significantly, increased in cortical slices from Cef-treated compared to saline-treated FHM2-KI mice ($162 \pm 5$ versus $145 \pm 5$ ms) (Fig 7C), and the frequency of spontaneous CSDs (2 out of 40 slices) was decreased in Cef-treated mice. The velocity of CSD propagation was similar in cortical slices from Cef- and saline-treated FHM2-KI mice ($3.82 \pm 0.08$ versus $3.83 \pm 0.07$) (Fig 7C). These findings indicate that Cef treatment rescues a small fraction of the facilitation of CSD induction in FHM2-KI mice without affecting the facilitation of CSD propagation.

Given the findings suggesting a necessary tight coupling between $\alpha_2$ NKA and GLT-1a in PAPs, we asked whether Cef

effectively increased the density of GLT-1a in cortical PAPs of W8887R/+ KI mice having 50% reduced $\alpha_2$ NKA expression. We used post-embedding EM to measure the density of GLT-1a gold particles associated with the membrane of PAPs in cortical sections from Cef- and saline-injected FHM2-KI mice. Interestingly, while the density of the membrane pool of GLT-1a in AxTs was larger in Cef-treated than in saline-treated FHM2-KI mice, the density of GLT-1a gold particles in the membrane of PAPs was similar in Cef- and saline-treated KI mice (Fig 8A and B, and Table 2). Accordingly, the GLT-1a protein expression level and the size of the GLT-1a$^+$ ir puncta that overlapped with VGLUT1 ir in the cortex of Cef- and saline-treated FHM2-KI mice were similar (Fig 8C and D). This is consistent with the Cef-induced increase in GLT-1a only at AxTs, given the relatively small fraction of GLT-1a in AxTs relative to that in PAPs (and the limit of resolution of confocal microscopy) (Chen *et al*, 2004; Melone *et al*, 2009). These findings provide an explanation for the relative inefficacy of Cef in the rescue of CSD facilitation and provide further support to the idea of a necessary tight coupling between $\alpha_2$ NKA and GLT-1a in PAPs. Given that Cef did not increase the density of GLT-1a in PAPs, the partial rescue of the facilitation of CSD induction in Cef-treated FHM2-KI mice may be due to the increased Glu reuptake in AxTs (due to higher GLT-1a expression) and/or to some other effect of Cef not related to GLT-1 expression. Mechanisms involving changes in expression of the $\alpha_2$ NKA or the glial Glu-cystine antiporter xCT (that appears to be coregulated with GLT-1 in nucleus accumbens after chronic treatment

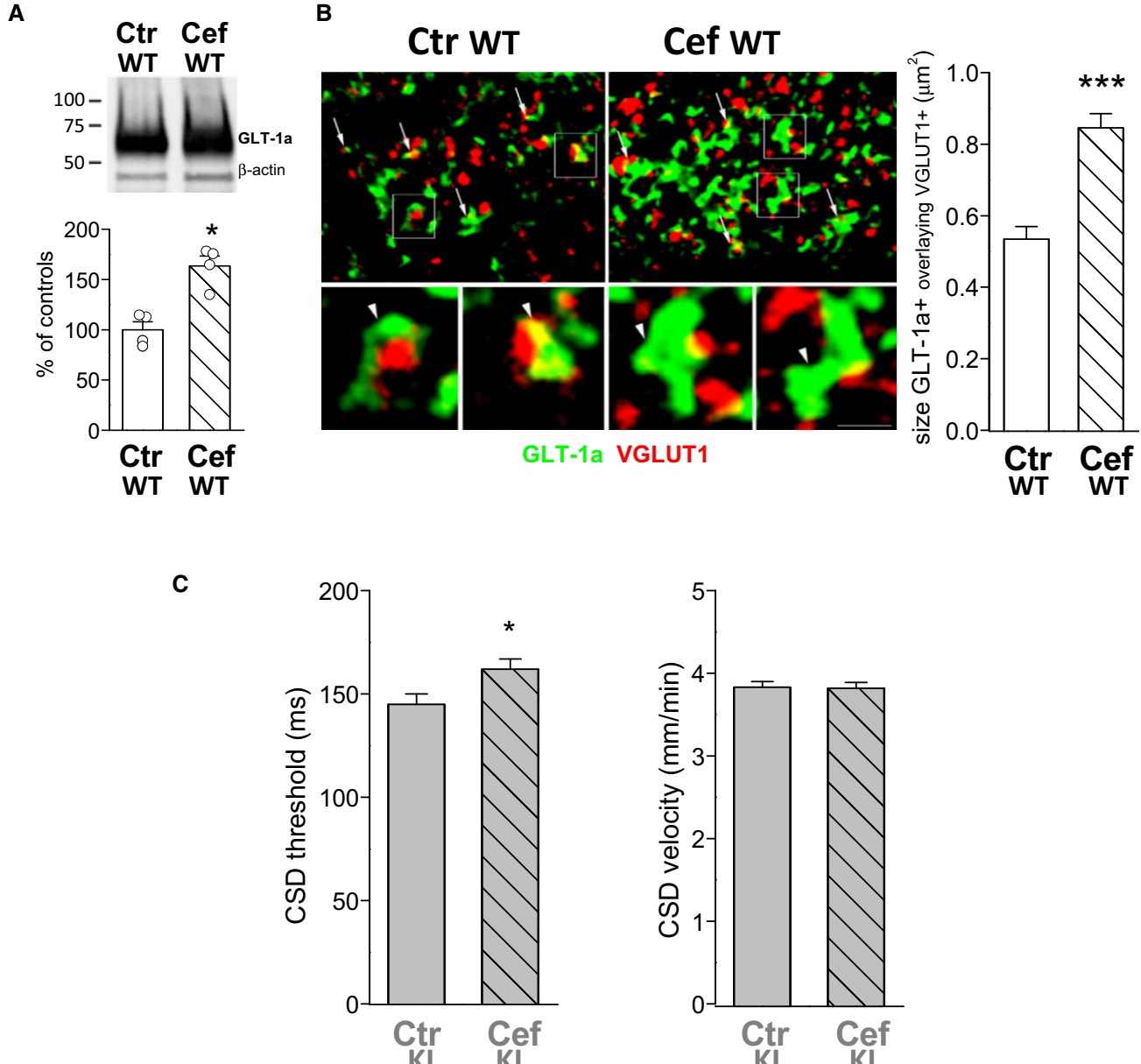

**Figure 7.  The CSD threshold is increased in cortical slices of FHM2-KI mice after Cef treatment that increases the GLT-1a expression in WT mice.**

A   Western blottings of GLT-1a in cortical crude synaptic membranes of P39 WT mice following Cef treatment for 8 days. GLT-1a levels are significantly increased in mice treated with Cef (Cef WT, $N$ = 4) compared to control saline-injected mice (Ctr WT, $N$ = 4) (Mann–Whitney $U$-test: *$P$ = 0.028). Data are mean ± SEM.

B   Visualization of GLT-1a[+] puncta (green) and VGLUT1[+] puncta (red) in sections from SI of P45-46 WT mice that were treated with saline (Ctr WT) or Cef (Cef WT). Arrows point to some GLT-1a/VGLUT1-related puncta. Framed regions (enlarged below) are examples of GLT-1a/VGLUT1-related puncta (arrowheads). Right, Cef treatment increased significantly the size of GLT-1a[+] puncta overlaying VGLUT1 (142 and 186 GLT-1a[+] puncta analyzed from 4 Ctr WT and 4 Cef WT mice, respectively; 3 sections/animal) (Mann–Whitney $U$-test: ***$P$ < 0.0001). Data are mean ± SEM. All microscopic fields are from layers II/III. Scale bar: 3.5 μm for left and right upper panels and 1 μm for enlarged framed areas.

C   CSD threshold and CSD velocity in cortical slices from P30-33 KI mice that were injected with saline (Ctr KI, $n$ = 38; $N$ = 7) or Cef (Cef KI, $n$ = 31; $N$ = 6). CSD threshold is 12% higher (Mann–Whitney $U$-test: *$P$ = 0.02) in Cef-treated compared to saline-treated KI mice. CSD velocity is not altered by Cef treatment ($P$ = 0.90). Data are mean ± SEM.

Source data are available online for this figure.

with cocaine or alcohol: (Knackstedt *et al*, 2010; Rao & Sari, 2014)) or Kir4.1 [the glial K$^+$ channel that plays a key role in K$^+$ spatial buffering: (Djukic *et al*, 2007; Kofuji & Newman, 2004)] do not seem to play a role, since the protein level of $\alpha_2$ NKA, xCT, and Kir4.1 was not affected by Cef treatment of FHM2-KI mice (Fig EV2).

As a second approach to study whether there is a causative relationship between reduced rate of Glu clearance by astrocytes and CSD facilitation, we investigated whether pharmacological reduction of the rate of Glu clearance in WT mice to a value similar to that in FHM2-KI mice reduces the threshold for CSD induction and increases the velocity of CSD propagation to values similar to those in the FHM2 mutants. To identify which concentration of drug produced a slowing of the rate of Glu clearance in WT mice close to that produced by the FHM2 mutation and obtain information on the time necessary to reach steady-state inhibition, we measured the STC in cortical slices from P22-23 WT mice before and after the application of different subsaturating concentrations of DL-TBOA. We identified a concentration of DL-TBOA (2.5 μM) that increased by 32 ± 2% the time constant of decay of the STC elicited by single pulse stimulation, thus producing a slowing of the rate of Glu clearance close to (although larger than) that produced by the FHM2 mutation (Fig 9A). In the presence of 2.5 μM DL-TBOA, the threshold for CSD induction was 36% lower than in control (142 ± 4 versus 220 ± 8 ms), and the velocity of CSD propagation was 20% higher (3.84 ± 0.09 versus 3.21 ± 0.08 mm/min) (Fig 9B). Thus, pharmacological inhibition of a fraction of GluTs does facilitate CSD induction and propagation in WT mice. In correlation with the larger slowing of the rate of Glu clearance produced by TBOA 2.5 relative to that produced by the FHM2 mutation (32 versus 21%: Figs 9A and 1C), the facilitation of CSD induction was also larger (36% versus 28% lower CSD threshold in TBOA 2.5 versus KI, respectively: Figs 9B and 6B), suggesting that the reduced rate of Glu clearance in FHM2-KI mice may account for a large fraction of the facilitation of CSD induction. To establish whether impaired Glu clearance in FHM2-KI mice may completely account for the facilitation of CSD induction, we measured CSD threshold and velocity in the presence of a concentration of DL-TBOA (1.5 μM) that produced a slowing of the rate of Glu clearance quantitatively similar to that produced by the FHM2 mutation (22 ± 3%: Fig 9C). In the presence of 1.5 μM DL-TBOA, the threshold for CSD induction was 23% lower than in control (170 ± 5 versus 220 ± 8 ms) and the velocity of CSD propagation was 13% higher than in control WT slices (3.61 ± 0.09 versus 3.21 ± 0.08) (Fig 9D). These data support the conclusion that the reduced rate of Glu clearance in FHM2-KI mice can account for most of the facilitation of CSD induction (82%, as estimated from the ratio of the relative reductions of CSD threshold produced by TBOA 1.5 and the FHM2 mutation: 23%/28%; cf also

similar CSD thresholds in WT TBOA 1.5 and FHM2 KI in Figs 9D and 6B), while it can account for only a fraction of the facilitation of CSD propagation in FHM2-KI mice (62%, as estimated from the ratio 13/21 of the relative increases in CSD velocity produced by TBOA 1.5 and the FHM2 mutation; cf also different CSD rates in WT TBOA 1.5 and FHM2 KI: unpaired *t*-test, *P* < 0.001).

## Discussion

We have studied the functional consequences of the reduced membrane expression of the $\alpha_2$ NKA pump in heterozygous W887R/+ FHM2-KI mice on the rate of Glu clearance by astrocytes during neuronal activity, as measured from the decay kinetics of the synaptically activated Glu transporter current recorded in cortical astrocytes. We have shown that Glu clearance by astrocytes during neuronal activity is slower in FHM2-KI compared to WT mice and that the density of GLT-1a in the membrane of astrocytic processes surrounding cortical excitatory synapses is about 50% reduced in the FHM2 mutants, a reduction that mirrors the reduced expression of the $\alpha_2$ NKA protein. Interestingly, the relative impairment of Glu clearance in FHM2-KI mice is activity dependent. In fact, when synaptic Glu release was induced by a train of pulses, the slowing down of the rate of Glu clearance in the FHM2 mutants was larger than that observed with a single pulse and was larger with a 100-Hz compared to 50-Hz train. This is consistent with and likely reflects the decreased Glu binding capacity of GluTs in the KI mice due to the reduced density of GLT-1a in PAPs. As a consequence, a relatively lower number of GluTs remain available to bind Glu at the end of the train in KI compared to WT astrocytes, resulting in a relatively larger slowing of the rate of Glu clearance.

These data provide the first direct experimental evidence, to our knowledge, for a key role of the $\alpha_2$ NKA pump in the clearance of synaptically released Glu during neuronal activity. Our findings suggest that this key role is based on a necessary tight coupling between the α2 pump and GLT-1 in cortical PAPs. These findings include (i) the strict colocalization of $\alpha_2$ NKA and GLT-1a in astrocytic processes in the vicinity of glutamatergic synapses; (ii) the 48% reduction in the membrane density of GLT-1a in KI PAPs, quantitatively similar to the reduction in the expression of $\alpha_2$ NKA, and, in contrast, the unaltered membrane density of GLT-1a in AxTs where $\alpha_2$ is not expressed; (iii) the insignificant increase in

**Figure 8.  The density of GLT-1a in the membrane of cortical perisynaptic astrocytic processes is not altered by ceftriaxone treatment in W887R/+ FHM2-KI mice, while the density of GLT-1a in the axon terminals is increased.**

A  Distribution of GLT-1a gold particles in PAP of saline-injected (Ctr KI, *N* = 4) and Cef-treated KI mice (Cef KI, *N* = 4) (P45-46). Cef treatment does not modify the density of total, cytoplasmic (arrows) and membrane-associated (arrowheads) gold particles in PAP of KI mice (see Table 2 for statistics).

B  Distribution of GLT-1a gold particles in AxT in SI of Ctr KI and Cef KI mice. The density of the membrane-associated gold particles (arrowheads) is increased in Cef-treated KI mice (see Table 2 for statistics). All microscopic fields in (A) and (B) are from layers II/III. *t*, total density; cyt, cytoplasmic density; mem, membrane density. Scale bar: 100 nm.

C  Western blottings of GLT-1a in cortical crude synaptic membranes of P39-KI mice following Cef treatment. GLT-1a levels are similar in Cef-treated (Cef KI, *N* = 4) and saline-injected (Ctr KI, *N* = 4) (Mann–Whitney *U*-test: *P* = 0.83). Data are mean ± SEM.

D  Visualization of GLT-1a$^+$ puncta (green) and VGLUT1$^+$ puncta (red) in KI mice that received saline (Ctr KI) and in Cef-treated KI mice (Cef KI) (P45-46). Arrows point to some GLT-1a/VGLUT1-related puncta; framed regions are examples of GLT-1a/VGLUT1-related puncta (arrowheads). Right, Cef treatment does not increase the size of GLT-1a$^+$ puncta overlaying with VGLUT1 (160 and 175 GLT-1a$^+$ puncta analyzed from the same 4 Ctr KI and 4 Cef KI used for post-embedding electron microscopy analysis; three sections/animal) (Mann–Whitney *U*-test: *P* = 0.31). Data are mean ± SEM. Scale bar: 3.5 μm for left and right upper panels and 1 μm for enlarged framed areas. All microscopic fields are from layers II/III.

Source data are available online for this figure.

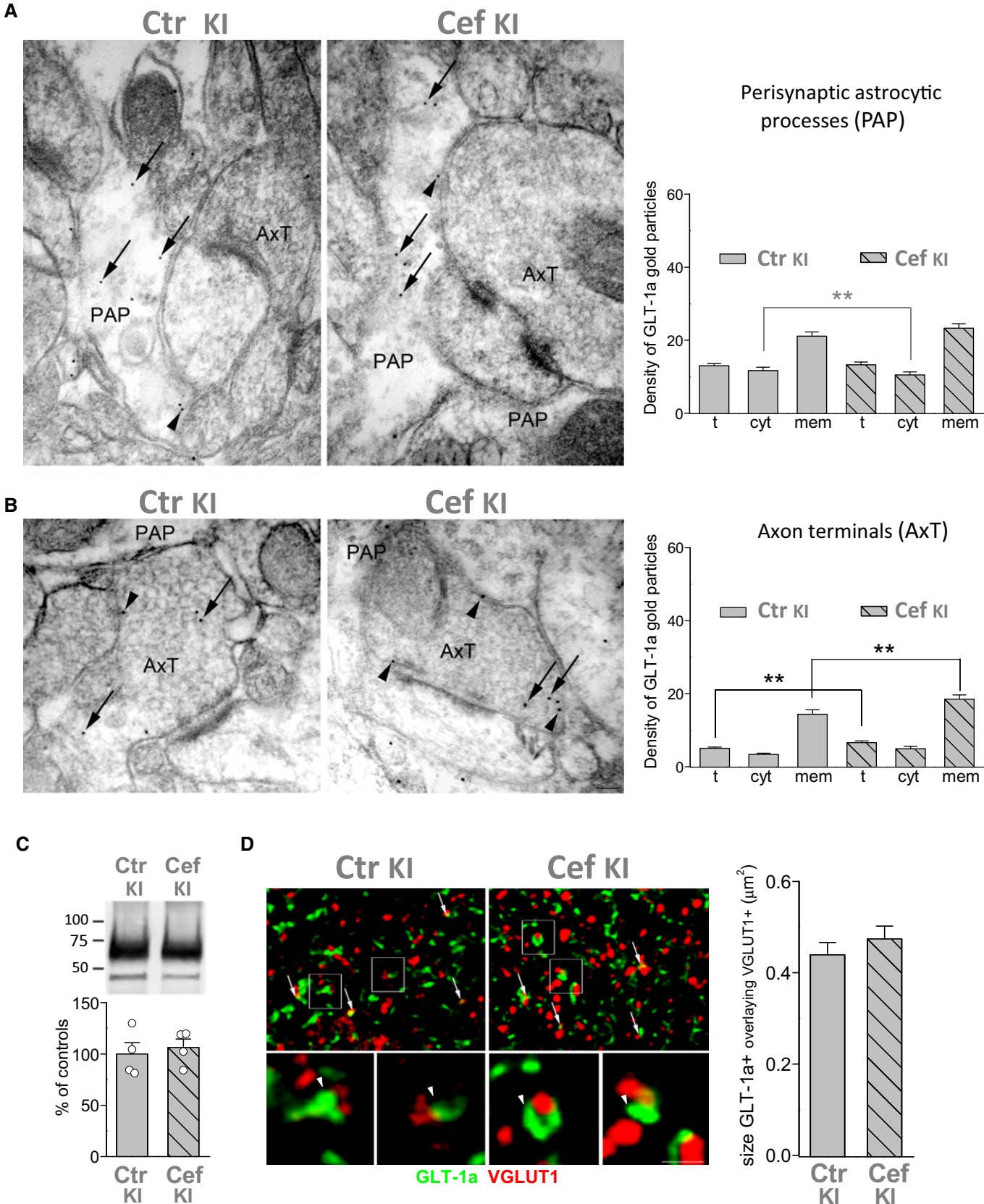

**Figure 8.**

Table 2.    **Density of GLT-1a gold particles in perisynaptic astrocytic processes and axon terminals of asymmetric synapses of saline (Ctr)- and ceftriaxone (Cef)-treated W887R/+ FHM2-KI mice.**

Density values are mean $\pm$ SEM; $n$ = the number of profiles. Comparison of densities between saline-treated ($N = 4$) and Cef-treated ($N = 4$) FHM2-KI mice was made using Mann–Whitney $U$-test.

| Localization | Ctr KI (particles/$\mu m^2$) | Cef KI (particles/$\mu m^2$) | Ctr versus Cef |
|---|---|---|---|
| Nucleus (background) | 0.51 $\pm$ 0.03 ($n = 25$) | 0.54 $\pm$ 0.01 ($n = 25$) | |
| Astrocytic processes | 12.98 $\pm$ 0.60 ($n = 354$) | 13.26 $\pm$ 0.71 ($n = 380$) | $P = 0.27$ |
| Plasma membrane | 21.13 $\pm$ 1.16 | 23.31 $\pm$ 1.19 | $P = 0.27$ |
| Cytoplasm | 11.70 $\pm$ 0.86 | 10.52 $\pm$ 0.83 | $P = 0.009$ |
| Axons terminal | 5.10 $\pm$ 0.30 ($n = 158$) | 6.56 $\pm$ 0.50 ($n = 166$) | $P = 0.002$ |
| Plasma membrane | 14.40 $\pm$ 1.22 | 18.50 $\pm$ 1.18 | $P = 0.007$ |
| Cytoplasm | 3.42 $\pm$ 0.29 | 4.85 $\pm$ 0.70 | $P = 0.068$ |

membrane density of GLT-1a in PAPs in Cef-treated FHM2-KI mice, in contrast with the increased density of GLT-1a in AxTs.

By using astrocytes as $[K]_e$ biosensors, we have also studied the functional consequences of the reduced expression of $\alpha_2$ NKA in heterozygous W887R/+ KI mice on the rate of $K^+$ clearance, as measured from the decay kinetics of the $[K^+]_e$-dependent slow inward current elicited in astrocytes by neuronal activity and the ensuing $K^+$ efflux. The finding that the rate of $K^+$ clearance after a train of pulses at 50 Hz is slower in FHM2-KI mice provides the first direct experimental support for an important role of the $\alpha_2$ NKA pump in $K^+$ clearance following neuronal activity.

Can impaired Glu and $K^+$ clearance during neuronal activity explain the facilitation of initiation and propagation of experimental CSD in the FHM2 mouse model and the episodic vulnerability to CSD ignition in FHM2? Despite important progress, the mechanisms underlying the initiation of experimental CSD remain incompletely understood and controversial (Pietrobon & Moskowitz, 2014). Experimental data and computational models support the ideas that an increase in $[K^+]_e$ above a critical value is a key initiating event and that generation of a net self-sustaining inward current and a regenerative local $K^+$ release are essential components of the positive feedback cycle that confers to CSD its all-or-none characteristics and causes a complete neuronal depolarization if the removal of $K^+$ from the interstitium does not keep pace with its release (Somjen, 2001; Pietrobon & Moskowitz, 2014). The ion channels involved in the generation of the net self-sustaining inward current and in the regenerative local $K^+$ release essential for CSD ignition remain incompletely understood, although there is strong pharmacological support for a key role of NMDARs (Pietrobon & Moskowitz, 2014). Investigation of the mechanisms underlying the facilitation of experimental CSD in FHM1-KI mice, carrying a gain-of-function mutation in neuronal $Ca_V2.1$ channels, revealed an enhanced excitatory synaptic transmission at cortical pyramidal cell synapses (Tottene *et al*, 2009; Vecchia *et al*, 2015) and showed a causative link between enhanced glutamatergic transmission at cortical synapses and facilitation of initiation and propagation of experimental CSD (Tottene *et al*, 2009).

Here, we investigated whether there is a causative relationship between the reduced rate of Glu clearance at cortical synapses and the facilitation of CSD in FHM2-KI mice, using two different approaches.

In the first approach, we investigated whether the facilitation of CSD in FHM2-KI mice could be rescued by systemic treatment with Cef, a drug that increased around 60% the expression of GLT-1a in neocortex of WT mice. Cef was able to rescue a small portion of the facilitation of CSD induction without affecting the facilitation of CSD propagation. The findings that neither the total expression of GLT-1a in the cortex nor the membrane density of GLT-1a in PAPs was significantly increased in Cef-treated FHM2-KI mice made it difficult to draw a clearcut conclusion regarding the role of impaired Glu clearance by astrocytes in CSD facilitation. The observation that Cef increased GLT-1a expression in AxTs in FHM2-KI mice suggests the possibility that the small increase in CSD threshold produced by Cef may be due to the increased reuptake of Glu in AxTs. If correct, this would be remarkable because GLT-1a in AxTs is only a small fraction of the total brain GLT-1a (Chen *et al*, 2004; Furness *et al*, 2008; Melone *et al*, 2009); accordingly, selective deletion of GLT-1a in neurons does not give rise to any apparent neurological abnormality in contrast with selective deletion in astrocytes (Petr *et al*, 2015). Alternatively, some other effect of Cef, not related to GLT-1 expression, might underlie its small effect on CSD threshold, although the unaltered protein levels of $\alpha_2$ NKA, xCT, and Kir4.1 in Cef-treated FHM2-KI mice make the involvement of mechanisms mediated by these proteins unlikely.

In the second approach, we investigated whether pharmacological reduction in the rate of Glu clearance by astrocytes in WT mice to values similar to those in FHM2-KI mice could lead to a similar facilitation of CSD induction and propagation. Pharmacological inhibition of a fraction of GluTs in WT mice did lower the threshold for CSD induction and increased the velocity of CSD propagation. The quantitative comparison between the changes in CSD threshold and velocity produced by the FHM2 mutation and by subsaturating concentrations of DL-TBOA supports the conclusion that the reduced rate of Glu clearance by astrocytes can account for most of the facilitation of CSD initiation in FHM2-KI mice, leaving little room for other contributing mechanisms. In contrast, impaired Glu clearance by astrocytes can account for only a (relatively large) fraction of the facilitation of CSD propagation, suggesting that other mechanisms contribute. The observation that also Cef treatment in FHM2-KI mice had a differential effect on CSD threshold and velocity appears consistent with the interpretation that increased reuptake of Glu in AxTs might account for the small rescue of facilitation of CSD induction produced by Cef.

The present data, together with the findings in FHM1-KI mice (Tottene *et al*, 2009), support a model of CSD initiation in which (i) excessive glutamatergic synaptic transmission and activation of

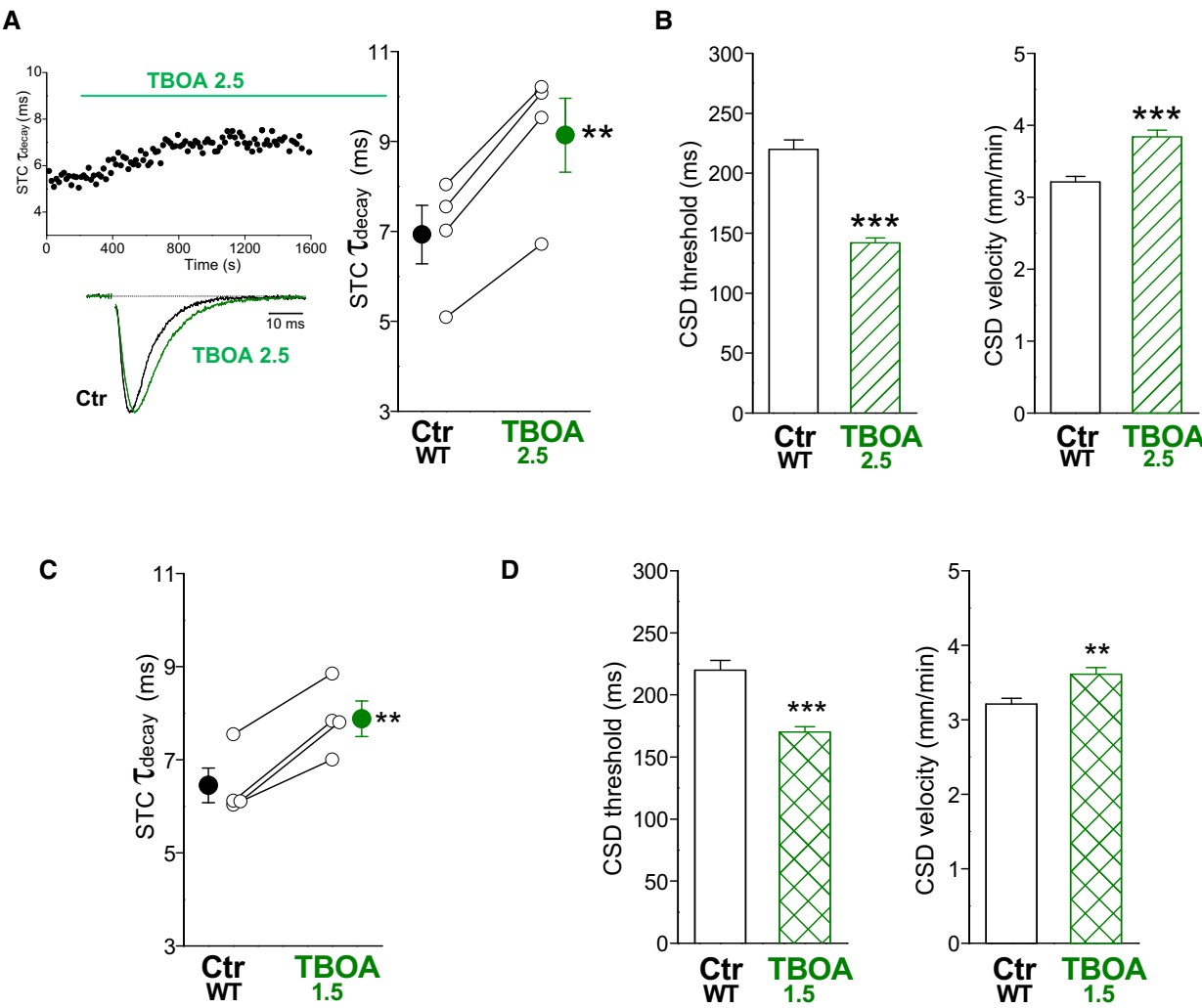

**Figure 9.  Facilitation of CSD induction and propagation after pharmacological inhibition of a fraction of glutamate transporters in WT mice.**

A   $\tau_{decay}$ of the STC evoked by single pulse stimulation in layer 1 astrocytes in acute cortical slices from P22-23 WT mice before (Ctr WT) and after the application of
2.5 μM DL-TBOA (TBOA 2.5) ($n = 4$; $N = 2$) (right panel). The traces on the left are the corresponding average normalized STCs, isolated as in Fig 1C. The STC $\tau_{decay}$ in
TBOA 2.5 is 32 ± 2% higher than in Ctr WT (paired $t$-test: **$P = 0.002$). The left top panel shows the time course of $\tau_{decay}$ of the transient component (due to the STC)
of the current recorded in an astrocyte during a representative experiment in which TBOA 2.5 was applied at the time indicated by the horizontal bar. The steady-
state effect was reached within 10–15 min from the beginning of the drug perfusion. Data are mean ± SEM.
B   CSD threshold and velocity measured in cortical slices from P22-23 WT mice after perfusion for 20 min with extracellular solution without (Ctr WT: $n = 23$; $N = 15$)
and with 2.5 μM DL-TBOA (TBOA 2.5: $n = 25$; $N = 8$). CSD threshold in TBOA 2.5 is 36% lower than in Ctr WT (Mann–Whitney $U$-test: ***$P < 0.0001$) and CSD velocity
is 20% higher (unpaired $t$-test: ***$P < 0.0001$). Data are mean ± SEM.
C   $\tau_{decay}$ of the STC evoked by single pulse stimulation in layer 1 astrocytes in acute cortical slices from P22-23 WT mice before (Ctr WT) and after the application of
1.5 μM DL-TBOA (TBOA 1.5) ($n = 4$; $N = 4$). The STC $\tau_{decay}$ in TBOA 1.5 is 22 ± 3% higher than in Ctr WT (paired $t$-test: **$P = 0.004$). Data are mean ± SEM.
D   CSD threshold and velocity measured in cortical slices from P22-23 WT mice after perfusion for 20 min with extracellular solution without (Ctr WT: $n = 23$; $N = 15$)
and with 1.5 μM DL-TBOA (TBOA 1.5: $n = 18$; $N = 6$). CSD threshold in TBOA 1.5 is 23% lower than in Ctr WT (Mann–Whitney $U$-test: ***$P < 0.0001$) and CSD velocity
is 13% higher (unpaired $t$-test: **$P = 0.003$). Data are mean ± SEM.

Source data are available online for this figure.

NMDARs are key elements in the positive feedback cycle that ignites CSD and (ii) the $\alpha_2$ NKA pumps exert a dampening role owing mainly to their key role in Glu clearance by astrocytes. There is evidence that cooperative activation of postsynaptic NMDARs by independent excitatory synapses due to Glu spillover occurs with, for example, high-frequency stimulation or by increasing the probability of Glu release or with pharmacological inhibition of GluTs, that is, in conditions in which the binding capacity of GluTs is overwhelmed by high extracellular Glu (Lozovaya et al, 1999; Arnth-Jensen et al, 2002; Tsukada et al, 2005). Most likely, cooperative activation of postsynaptic NMDARs also occurs with the experimental stimuli that ignite CSD, since they induce a large release of Glu (Pietrobon & Moskowitz, 2014; Enger et al, 2015). Within the framework of the proposed CSD model, one predicts a lower threshold for CSD induction in FHM2-KI compared to WT mice because in the FHM2 mutants depolarizing stimuli of lower intensity will release

enough Glu to overwhelm the binding capacity of GluTs and lead to cooperative activation of a sufficient number of NMDARs to initiate the positive feedback cycle that ignites CSD. The findings discussed above indicate that the contribution of the reduced rate of $K^+$ clearance by astrocytes to the facilitation of CSD initiation in FHM2-KI mice, if present, is quite small. This would be consistent with a more relevant role of the neuronal $\alpha_3$ NKA relative to the glial $\alpha_2$ NKA in $K^+$ buffering during the CSD-inducing stimuli. Evidence that this is the case during the CSD depolarization is provided by reports that the duration of the CSD depolarization is increased in heterozygous KI mice carrying a mutation that reduces the activity of $\alpha_3$ NKA (Hunanyan et al, 2015), whereas, in contrast, the CSD duration is not affected in FHM2-KI mice with reduced $\alpha_2$ NKA (Leo et al, 2011).

The typical slow rate of CSD propagation implies that it is mediated by diffusion of a chemical substance. Although it has long been debated whether the diffusing substance is $K^+$ or Glu, most evidence points to the diffusion of $K^+$ released into the interstitial space during CSD as the underlying mechanism (Pietrobon & Moskowitz, 2014; Enger et al, 2015). The very fast rate of Glu clearance compared to that of $K^+$ clearance is consistent with and supports this notion. The faster rate of CSD propagation in FHM2-KI mice is consistent with the idea that CSD propagation is mediated by $K^+$ diffusion because, in the framework of our CSD model, one predicts that in KI mice the $[K^+]_e$-dependent Glu release necessary to activate a number of NMDARs sufficient for ignition of CSD in contiguous regions is obtained when the propagating $K^+$ ions reach a lower concentration than in WT mice. Moreover, it is plausible to expect that the reduced rate of $K^+$ clearance by astrocytes in FHM2-KI mice increases the rate of CSD spread and thus contributes to the facilitation of CSD propagation.

Since activity-dependent intracellular $Na^+$ transients in hippocampal and cortical astrocytes are predominantly due to $Na^+$ influx mediated by GluTs (Chatton et al, 2000; Langer & Rose, 2009; Lamy & Chatton, 2011), and since in our FHM2-KI mice the reduced expression of $\alpha_2$ NKA is accompanied by a similar reduction in GLT-1 in PAPs, it remains unclear whether and to what extent activity-dependent $Na^+$ transients are altered in the astrocytes of FHM2-KI mice. However, if the reduced $\alpha_2$ NKA expression in FHM2-KI mice significantly affects $Na^+$ homeostasis in cortical astrocytes, other possible hypothetical mechanisms that might contribute to CSD facilitation (to be possibly investigated in future studies) could result from slowing (or even reversal) of $Na^+$-dependent transporters such as the electrogenic $Na^+/HCO_3^-$ cotransporters or the $Na^+/Ca^{2+}$ exchangers (Kirischuk et al, 2012; Rose & Karus, 2013) (see Discussion in Appendix).

Both FHM2 and FHM1 mouse models are characterized by excessive glutamatergic transmission, due to either increased Glu release (FHM1) or reduced Glu clearance (FHM2). In contrast with the enhanced excitatory synaptic transmission at pyramidal cell synapses, inhibitory transmission at fast-spiking and other multipolar interneuron synapses was unaltered in FHM1-KI mice (Tottene et al, 2009; Vecchia et al, 2014, 2015). Interestingly, in the cortex, the $\alpha_2$ NKA pump is localized in astrocytic processes surrounding glutamatergic synapses, but is not present in astrocytic processes surrounding GABAergic synapses (Cholet et al, 2002), suggesting that FHM2 mutations likely affect excitatory but not inhibitory synaptic transmission. The differential effect of FHM1 and FHM2

mutations on excitatory and inhibitory synaptic transmission implies that, most likely, the neuronal circuits that dynamically maintain a tight balance between excitation and inhibition during cortical activity are functionally altered in FHM1 and FHM2, and suggests that dysfunctional regulation of the cortical excitatory inhibitory balance (E/I) may be a common feature of the FHM brain. This supports the view of migraine as a disorder of brain excitability characterized by dysfunctional regulation of the cortical E/I balance (Vecchia & Pietrobon, 2012; Ferrari et al, 2015) and gives insights into possible mechanisms underlying the susceptibility to ignition of "spontaneous" CSDs in FHM, and possibly migraine. It seems plausible to hypothesize that excessive glutamatergic transmission and dysfunctional regulation of the E/I balance in FHM may in certain conditions (e.g., with sensory overload or with other migraine triggers) lead to overexcitation and network hyperactivity with the consequent excessive $K^+$ increase and NMDAR activation (that leads to further $K^+$ increase), thus creating the conditions for initiation of the positive feedback cycle that ignites CSD. Similar mechanisms may underlie the susceptibility to CSD ignition in common forms of migraine for which there is indirect evidence consistent with enhanced cortical excitatory transmission (Pietrobon & Moskowitz, 2013) and for which genome-wide association studies have identified susceptibility loci potentially leading to enhanced glutamatergic transmission and/or dysregulated brain E/I balance (Anttila et al, 2010, 2013; Chasman et al, 2011; Freilinger et al, 2012).

# Materials and Methods

### Animals

Experiments were performed using heterozygous knockin (KI) mice harboring the W887R FHM2 mutation ($Atp1a2^{+/R887}$ mice: Leo et al, 2011) and their wild-type (WT) littermates (background C57BL6J, male and female in equal or near equal number). Animals were housed in specific pathogen free conditions, maintained on a 12-h light/dark cycle, with free access to food and water. Animals were anesthetized with isoflurane or an intraperitoneal (i.p.) injection of chloral hydrate (300 mg/kg) and the brains removed for acute slices preparation or prepared for immunocytochemistry and Western blotting studies. For Cef studies, mice received a daily i.p. injection (8–10 A.M) of saline or Cef (Rocefin, Roche S.p.A., Milano, Italy; 200 mg/kg/day dissolved in saline) for 7–8 days. Twenty-four hours after the final injection, the animals were anesthetized and brains removed. All experimental procedures involving animals and their care were carried out in accordance with National laws and policies (D.L. n. 26, March 14, 2014) and with the guidelines established by the European Community Council Directive (2010/63/UE) and were approved by the local authority veterinary services.

### Statistics

After assessing for normal distribution (using the Shapiro–Wilk or the Kolmogorov–Smirnov test), comparison between two groups was made using two-tailed unpaired or paired t-test for normal distributed data or the Mann–Whitney U-test for nonparametric data. Equal variances were assumed. Data are given as mean ± SEM; differences were considered statistically significant if

$P < 0.05$ (*$P < 0.05$; **$P < 0.01$; ***$P < 0.001$). No statistical methods were used to choose sample sizes that were estimated based on previous experience and are in line with those in the literature. No animals were excluded from the analysis.

### Electrophysiology

*Whole-cell patch-clamp recordings from astrocytes in acute brain slices*
Acute coronal slices of the somatosensory cortex were prepared from P22-23 mice (unless otherwise specified) of either sex as described in Tottene et al (2009) and in Appendix Supplementary Methods. Patch-clamp recordings were made following the standard techniques. Brain slices were continuously perfused in a submersion chamber with fresh extracellular solution at 30°C at a flow rate of 3 ml/min. Recordings were made from layer 1 astrocytes deeper than 45 μm from the slice surface. The cell bodies of astrocytes were visualized using an upright microscope equipped with infrared light and DIC optics and identified by their small soma size (< 10 μm), low input resistance (< 20 MΩ; WT: $13.2 \pm 0.7$ MΩ, $n = 42$; KI: $14.2 \pm 1.2$ MΩ, $n = 24$; $P = 0.42$), very negative resting membrane potential (WT: $-91 \pm 0.6$ mV, $n = 42$; KI: $-91 \pm 0.6$ mV, $n = 24$, after LJP correction of $-10$ mV), inability to generate action potentials and linear current–voltage relationships, typical of so-called passive astrocytes (Bergles & Jahr, 1997; Bernardinelli & Chatton, 2008) (Scimemi & Diamond, 2013).

Astrocyte internal solution contained (in mM) 115 K-gluconate, 6 KCl, 4 MgATP, 0.3 NaGTP, 10 Na-phosphocreatine, 10 HEPES, 5 glucose (pH 7.25 with KOH, osmolarity 295 mOsm with sucrose). The extracellular solution contained 125 mM NaCl, 2.5 mM KCl, 1 mM MgCl₂, 1 mM CaCl₂, 25 mM NaHCO₃, 1.25 mM NaH₂PO₄, 25 mM glucose, saturated with 95% O₂ and 5% CO₂. For the recording of Glu transporter-mediated currents, the extracellular solution also contained antagonists of AMPA receptors (10 μM NBQX), NMDA receptors (NMDARs; 50 μM D-AP5 and 20 μM (+)-MK801), and GABA$_A$ receptors (20 μM (+)-bicuculline or 5 μM gabazine) to block neuronal postsynaptic current flow (Bergles & Jahr, 1997).

Currents were evoked in astrocytes (held close to the resting potential at -90 mV, after LJP correction) by passing constant current pulses (100 μA, 100 μs) every 15 s (or 20 s with train of pulses) through a concentric bipolar tungsten electrode (TM33CCINS, World precision Instruments, Inc., Sarasota, FL, USA) placed in layer 1 at least 200 μm away from the recorded astrocyte (Fig 1A). Access resistance was monitored continuously throughout the experiments and was typically < 20 MΩ (without compensation); experiments where it changed more than 20% (or had access resistance > 25 MΩ) were excluded from the data. The recorded WT and KI astrocytes had similar access resistance (WT: $14.7 \pm 0.6$ MΩ, $n = 66$; KI: $16.0 \pm 0.8$ MΩ, $n = 47$; $P = 0.15$). Pipette resistance was 5–6 MΩ. Currents were sampled at 10 kHz and filtered at 2 kHz.

*Analysis of patch-clamp recordings*
Synaptically activated Glu transporter currents (STCs) elicited by single pulse stimulation were isolated pharmacologically using saturating concentrations of the GluTs inhibitor (3S)-3-[[3-[[4-(trifluoromethyl)benzoyl]amino]phenyl]methoxy]-L-aspartic acid (TFB-TBOA; Tocris Cookson Ltd., Bristol, UK) (0.5–15 μM). The STC was obtained by subtracting the residual current remaining in the presence of TFB-TBOA (TBOA) from the total control current

(Scimemi & Diamond, 2013). To obtain a measure of the rate of clearance of synaptically released Glu by cortical astrocytes, the decay of the STC was best fitted by a single exponential function (Diamond, 2005). STCs elicited by single pulse stimulation were also isolated nonpharmacologically by using an exponential waveform approximating the average TBOA-insensitive current (Diamond, 2005; Devaraju et al, 2013; Scimemi & Diamond, 2013). The average normalized residual current recorded in the presence of TBOA was obtained by pooling the normalized TBOA-insensitive currents recorded in 16 slices (recordings from 7 WT and 9 FHM2 astrocytes were pooled together because the time course of the residual current in WT and KI slices was similar). The rising phase of the average normalized TBOA-insensitive current was approximated by a monoexponential function (1-exp $(-t/\tau_{\text{rise}})$) with $\tau_{\text{rise}} = 2.35$ ms (Fig 1C). The STC was obtained by subtracting from the total current elicited in astrocytes an exponential function A(1-exp $(-t/\tau_{\text{rise}})$) with $\tau_{\text{rise}} = 2.35$ ms and with A equal to the maximal current measured in individual astrocytes at about 60 ms after stimulation when the transient transporter current had decayed to zero (Diamond, 2005; Devaraju et al, 2013; Scimemi & Diamond, 2013). When STCs were elicited by trains of 10 pulses at high frequency (50 or 100 Hz), it was not possible to isolate them pharmacologically (by subtracting the residual current in the presence of TBOA from the control current elicited by the train) because in the presence of TBOA the residual current measured after the train was larger than in control. Therefore, the STC elicited by the 10$^{\text{th}}$ pulse in the train was obtained from experiments in which single stimuli were alternated with trains of 9 and 10 pulses at 50 or 100 Hz (Diamond & Jahr, 2000) as follows. First, the current elicited by 9 pulses was subtracted from that elicited by 10 pulses (Fig 2A, left) to obtain the 10-9 pulses difference current (trace a-b); then, the STC elicited by the 10$^{\text{th}}$ pulse (Fig 2A, right: trace a-b-c) was obtained in isolation by subtracting the exponential function that simulates the TBOA-insensitive current elicited by a single pulse (trace c, obtained as explained above) from the 10-9 pulses difference current (trace a-b) (Fig 2A, right). This could be done because the amplitude of the TBOA-insensitive current elicited by a single pulse was similar to the amplitude of the steady component of the 10-9 pulses difference current (e.g. WT: $23 \pm 3$ pA versus $21 \pm 3$ pA with 50-Hz train, $n = 19$; $21 \pm 3$ pA versus $20 \pm 2$ pA with 100-Hz train, $n = 14$; similar finding for KI: $25 \pm 6$ pA versus $24 \pm 5$ pA with 50-Hz train, $n = 15$; $32 \pm 7$ pA versus $27 \pm 5$ pA with 100-Hz train, $n = 11$).

Sweeps are averages of at least five responses. Stimulation artifacts have been truncated.

### Cortical spreading depression

Cortical spreading depression was elicited and measured in acute coronal slices of the somatosensory cortex of WT and FHM2-KI mice as in Tottene et al (2009), but most recordings were at 30°C rather than at room temperature and the rate of perfusion of the slices was higher (6 ml/min, unless otherwise specified). See Appendix Supplementary Methods for details.

### Immunocytochemical studies

Mice were anesthetized with an i.p. injection of chloral hydrate (300 mg/kg) and perfused transcardially with a flush of saline

solution followed by 4% freshly depolymerized paraformaldehyde in 0.1 M phosphate buffer (PB; pH 7.4). Brains were removed, post-fixed in the same fixative (4 weeks for both double-labeling immunofluorescence and immunogold studies; 2 h for triple-labeling immunofluorescence observations), and cut in coronal sections on a vibratome in 50-μm sections that were collected in PB until processing (Melone *et al*, 2009).

*Immunofluorescence*

GLT-1a/VGLUT1 double-labeling studies were performed in sections from WT and KI mice and then, in a second set of experiments, in sections from saline- or Cef-treated WT mice and saline- or Cef-treated KI mice. To minimize the effects of procedural variables, GLT-1a/VGLUT1 double-labeling staining of sections from WT and KI groups and then of WT saline, WT Cef, KI saline, and KI Cef groups was performed in parallel using well-characterized antibodies (Rothstein *et al*, 1994; Melone *et al*, 2005; Omrani *et al*, 2009) and standard procedures as described in Melone *et al* (2009) and detailed in Appendix Supplementary Methods.

*Confocal microscopy and data analysis*

Collection and data analysis of confocal microscopic fields were performed by a blinded observer. Procedures were as described previously (Melone *et al*, 2005, 2009; Bragina *et al*, 2006) and are detailed in Appendix Supplementary Methods.

*Immunogold*

Sections from WT and KI animals and from KI mice that received saline or Cef were processed for the osmium-free method (Phend *et al*, 1995). Procedures were as described previously (Melone *et al*, 2009) and are detailed in Appendix Supplementary Methods. To minimize the effects of procedural variables, post-embedding procedure of grids from WT and KI groups and then from KI saline and KI Cef groups was performed in parallel.

*Electron microscopy and data analysis*

Collection and analysis of electron microscopic fields were performed by a blinded observer as described (Melone *et al*, 2009) and detailed in Appendix Supplementary Methods.

*Western blotting*

For western blotting studies, mice were anesthetized with chloral hydrate (300 mg/kg i.p.) and decapitated. Cerebral cortex was quickly separated, tissue was homogenized (Melone *et al*, 2001), and crude synaptic membranes were prepared (Danbolt *et al*, 1990). To minimize the procedural variables, homogenates from Cef and saline mice were loaded onto the same gel and in a blinded manner. Procedures were as described (Bragina *et al*, 2006) and detailed in Appendix Supplementary Methods, together with antibodies and controls.

**Expanded View** for this article is available online.

## Acknowledgements

We thank Michele Bellesi for help with Western blotting. This work was supported by Telethon Italy Grant GGP14234 (to D.P.), the Italian Ministry of University and Research (PRIN2010) (to D.P. and F.C.), the University of Padova

---

**The paper explained**

**Problem**

Migraine is a common disabling brain disease. Cortical spreading depression (CSD) is thought to play a key role in migraine pathogenesis in that it underlies migraine aura and may trigger migraine headache. The mechanisms of the primary brain dysfunction underlying the susceptibility to CSD ignition in the human brain and the onset of a migraine attack remain largely unknown and are a major open issue in the pathophysiology of migraine. To gain insights into this question, we studied the mechanisms underlying facilitation of CSD in a knockin (KI) mouse model of familial hemiplegic migraine type 2 (FHM2), a rare monogenic form of migraine with aura caused by loss-of-function mutations in the gene encoding the $\alpha_2$ subunit of the $Na^+$, $K^+$ ATPase ($\alpha_2$ NKA), an isoform almost exclusively expressed in astrocytes in the adult brain.

**Results**

Combining patch-clamp recordings from astrocytes in acute cortical slices with immunohistochemistry and immunogold electron microscopy, we show that (i) the rates of glutamate and $K^+$ clearance by cortical astrocytes during neuronal activity are reduced in heterozygous KI mice that carry the W887R FHM2 mutation and have 50% reduced $\alpha_2$ NKA protein in the brain, thus providing direct evidence for a key role of $\alpha_2$ NKA in clearance of synaptically released glutamate and in extracellular $K^+$ buffering during neuronal activity and (ii) the density of GLT-1a glutamate transporters in the membrane of astrocytic processes surrounding cortical excitatory synapses is about 50% reduced in the FHM2 mouse model, pointing to a specific tight coupling between $\alpha_2$ NKA and GLT-1a in the astrocytic processes. Measurements of CSD threshold and velocity in cortical slices from FHM2-KI mice treated with ceftriaxone and from wild-type mice after pharmacological inhibition of a fraction of glutamate transporters provide evidence that the defective glutamate clearance by astrocytes underlies most of the facilitation of CSD initiation (and a large fraction of the facilitation of CSD propagation) in the FHM2 mouse model.

**Impact**

By uncovering the key mechanism underlying the facilitation of CSD in the FHM2 mouse model, this study moves forward our understanding of the molecular and cellular mechanisms that may underlie the brain susceptibility to CSD ignition in migraine and point out a direction in which to search for novel migraine therapies. In particular, it points to excessive cortical glutamatergic synaptic transmission as a key mechanism underlying vulnerability to CSD ignition in migraine and to the enhancement of astrocytic glutamate transporter function as a possible new therapeutic strategy.

(Progetto Ateneo 2012) (to D.P.), intramural funds from INRCA IRCCS (to F.C.) and Universita' Politecnica delle Marche (to F.C.).

## Author contributions

CC, MM, and AT performed the experiments, analyzed the data, discussed interpretation of the data, and revised critically the manuscript. LB and GC contributed to the acquisition, analysis, and discussion of the data. MS and GC contributed to the experimental design, discussed interpretation of the data, and revised critically the manuscript. GC also provided the FHM2 mouse model. FC and DP designed the study, supervised the experiments, and analyzed and interpreted the data. DP conceived the study and wrote the paper.

## Conflict of interest

The authors declare that they have no conflict of interest.

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
