## [Review Process File · EMBO Molecular Medicine]

Defective glutamate and K⁺ clearance by cortical astrocytes in familial hemiplegic migraine type 2

Clizia Capuani, Marcello Melone, Angelita Tottene, Luca Bragina, Giovanna Crivellaro, Mirko Santello, Giorgio Casari, Fiorenzo Conti and Daniela Pietrobon

Corresponding author: Daniela Pietrobon, University of Padova and CNR Institute of Neuroscience

Review timeline:

Submission date:	15 October 2015
Editorial Decision:	17 November 2015
Revision received:	29 March 2016
Editorial Decision:	17 May 2016
Revision received:	27 May 2016
Accepted:	31 May 2016

Editor: Céline Carret

Transaction Report:

1st Editorial Decision

17 November 2015

Thank you for the submission of your manuscript to EMBO Molecular Medicine. We have now heard back from the three referees whom we asked to evaluate your manuscript. Although the referees find the study to be of potential interest and rather well performed, they also raise a number of concerns that need to be addressed in the next final version of your article.

You will see from the comments pasted below, that all three referees have somehow overlapping concerns regarding functional insights and request additional experiments and clarifications / explanations when mentioned, in order to provide some mechanistic understanding and thereby improve impact and robustness of the data.

Given these evaluations, I would like to give you the opportunity to revise your manuscript, with the understanding that the referee concerns must be fully addressed and that acceptance of the manuscript would entail a second round of review. Please note that that it is EMBO Press's policy to allow only a single round of revision, and that acceptance or rejection of the manuscript will therefore depend on the completeness of your response and the satisfaction of the referees with it. Revisions should be resubmitted within 3 months unless otherwise arranged with the editor.

I look forward to seeing a revised form of your manuscript.

**** Reviewer's comments ****

Referee #1 (Comments on Novelty/Model System):

Suggestions for improving the quality/impact of the study and model:

- better description of the glutamate transporter current analysis
- addressing effect of altered Na⁺ fluxes for CSD susceptibility in FHM2 mice
- assessment of glutamate and K⁺ concentration in vivo in FHM2 mice in relation to high frequency stimulation and CSD

Referee #1 (Remarks):

In their manuscript the authors aimed to investigate mechanisms underlying the increased susceptibility to cortical spreading depression (CSD) in heterozygous FHM2 knock-in (KI) mice. This is a clinically relevant question since it supports the hypothesis that FHM2 mutations cause impairment of astrocytic glutamate clearance. The authors use advanced electrophysiological recording and analysis techniques to assess changes in astrocytic function in cortical slices from young (3-5 week old) FHM2 heterozygous KI mice. The authors report a reduction in astrocytic glutamate transporter current and reduced density of GLT-1a glutamate transporters in perisynaptic astrocytic processes. The observations correlated with an enhanced susceptibility to KCl-evoked CSD in the slices. Although systemic treatment with ceftriaxone caused enhancement of astrocytic GLT-1a expression in WT mice, in FHM2 mice ceftriaxone treatment did not normalize the CSD threshold values; which was in line with the observation that ceftriaxone treatment in FHM1 mice did not alter the astrocytic membrane-bound expression of GLT-1. Lastly, the authors demonstrated a reduced astrocytic slow decaying inward current likely reflecting reduced astrocytic K⁺ influx in slices from FHM2 mice.

The work is well written and all studies appear to have followed sound methods and analytical approaches. The manuscript in my opinion contains two major shortcomings. The authors overemphasize the observed effects of the FHM2 mutation on glutamate transporter function as a key mechanism underlying CSD susceptibility, while not stressing a role for the observed reduction in astrocytic K⁺ conductance. Further, effects of the FHM2 mutation on astrocytic Na⁺ homeostasis is not discussed, while this is not unlikely to influence CSD susceptibility as well.

Major:

1. The authors overemphasize their findings on the reduced astrocytic glutamate transporter current and the effects of ceftriaxone on CSD susceptibility. Also, the calculation of the glutamate transporter current, which is one of the key readouts for FHM2 astrocytic function in this manuscript, needs to be better described.

- i) Throughout the paper, the authors often refer directly to their experimental data as 'reduced glutamate clearance', while this is not what was actually measured. E.g. in the abstract line 27 'we show reduced rates of glutamate clearance' should state more precisely e.g. 'we showed reduced glutamate transporter currents, indicating reduced rates of glutamate clearance'. It would have been nice to have direct evidence for enhanced glutamate and K⁺ levels e.g. via glutamate or pH sensor measurements.
- ii) The conclusions in the manuscript should be rephrased to be in line with the experimental data, that a) did not show an effect of ceftriaxone on astrocytic GLT-1 expression, b) show only a minor increase in CSD threshold upon ceftriaxone, without a change in CSD velocity, and c) indicated a reduced rate of K⁺ clearance. For example, the conclusion of the Abstract the statement 'we obtain evidence that defective glutamate clearance contributes to facilitation of CSD initiation in FHM2,....' needs rephrasing in this respect, as well as the conclusions in the Discussion.
- iii) The calculation of the synaptically-activated transporter current STC that is used throughout the paper and depicted first in Figure 1A and 1B, needs to be described better in the Methods and Results section and Figure legend. In the Methods on p.25, line 615, it is not clear which data are pooled. Were the 16 experiments from 16 slices of WT mice, or a pool of WT and KI recordings? To aid the understanding for the non-specialized readership, I suggest to add a schematic of the recording paradigm, indicating the site of stimulation and recording of the synaptically-activated glutamate transporter current STC. Such a schematic will also make clear that the STC data are interpreted as measure of astrocytic glutamate clearance.

2. The effect of the glial $\alpha 2$ Na⁺/K⁺ ATPase in reducing the intra-astrocytic sodium concentration (by Na⁺ efflux coupled to K⁺ influx) is not addressed, while it could also contribute to the enhanced CSD susceptibility in FHM2 mice. A predicted reduction in Na⁺/K⁺ ATPase function in FHM2 KI mice would be expected to influence astrocyte transporter mechanisms relying on Na⁺ influx. For example, apart from Na⁺ dependent glutamate uptake by astrocytes, glial Na⁺ influx is directly coupled to H⁺ extrusion and HCO₃⁻ uptake, via NHE and NBC transporters. Reduced activity of these transporters that may be expected with reduction in Na⁺/K⁺ ATPase function in FHM2 KI mice could alter cortical pH which, in case of a more alkaline environment, would enhance neuronal excitability and can influence CSD susceptibility. These aspects need be addressed in the discussion on the CSD susceptibility experiments in cortical slices from FHM2 mice.

Minor:

- a) The authors have not specified when n-numbers refer to number of slices or number of mice. I suggest to uniformly depict by 'n' the number of slices, and indicate with 'N' the number of mice that were used for those slices.
- b) Ages of mice should be mentioned for each of the subfigures, e.g. age is not mentioned for the experiments from Figure 1 and Figure 6A.
- c) Methods: it is not clearly described whether electrophysiological studies and analyses were performed by a blinded observer, as was mentioned for the immunocytochemical studies.
- c) Results: the authors should give information on the astrocytic input resistance for WT compared to KI recordings, and mention # exclusions (as indicated in the Methods) based on variable or too high access resistance, to show that WT and KI astrocytes had comparable access resistance.
- d) Results Figure 5A: the delay from KCl puff to CSD appears longer for FHM1 KI compared to WT mice. This parameters needs to be mentioned in the
- d) Discussion: given the described neuronal expression of $\alpha 2$ - Na⁺/K⁺ ATPase during embryonic development, possible effects of the FHM2 mutation on cortical synaptic function in adulthood, e.g. as a result of alterations in cortical network development, should be addressed.

Referee #2 (Remarks):

This is a very interesting, carefully executed, well-written and potentially important paper. It focuses on the familial hemiplegic migraine type 2 mutation that is caused by loss of function mutations in the $\alpha 2$ sodium-potassium ATPase (NKA). The present work demonstrates by a combination of physiology and anatomical studies a loss of glutamate clearance due to decreased expression of the glutamate transporter GLT-1, which physically interacts with the NKA, as well as a decrease in potassium clearance from the extracellular space. Presumably GLT-1 function is dependent upon having a NKA close by, and the expression of GLT-1 is in some way dependent upon the expression of NKA. The consequences of these deficits appears to be a facilitation of chronic spreading depression, which underlies migraine aura and possibly migraine itself. The recognition that GLT-1a is expressed in axon terminals, and the demonstration that its expression in this location is not changed in the KI, as expected given that $\alpha 2$ subunit is astrocytic and not neuronal, is very nice. Also, the authors have shown, probably for the first time, that ceftriaxone increases GLT-1a protein expression in axon terminals. The central issue here is whether the facilitation of CSD is due to the glutamate clearance abnormality or the K clearance abnormality--or some other abnormality, an issue that remains unresolved.

There are some problems that need to be addressed:

1. A primary observation is that there is a 20% reduction in the mean size of GLT-1a + puncta in the FHM2 KI mice. Does this mean that the size of spines is smaller? It is not clear that a decrease in protein expression of GLT-1a would make for a decrease in size of the structures that express GLT-1a. A change in spine size might be important for the pathophysiology of CSD.
2. The authors observe that with increasing temperature, the threshold for CSD induction decreases. Since the activity of glutamate transporters has a Q₁₀ of 2-3, these data re temperature effect seem

hard to rationalize. Perhaps the writing is just confusing; compare what is stated on p. 12 with the last sentence of same paragraph.

3. The authors observed, with ceftriaxone, a 50% increase in protein on western blots. Some labs have not observed an effect of cef in normal animals. These data need to be normalized to an internal standard.
4. Cef seems to increase the size of puncta without increasing the density of GLT-1a gold particles. This is hard to understand in view of: 1) the increase on western blot; 2) the change in size of GLT-1a puncta, if, indeed, the size of GLT-1a puncta reflect anything about GLT-1a expression, and not a change in size of something else, like spine size.
5. The authors suggest that the partial rescue of the facilitation of CSD in cef treated animals is likely due to increased reuptake in terminals, because there is a change in density of GLT-1a immunogold in terminals but not in puncta. But, this is hard to believe, given that only 5-10% of GLT-1a is expressed in terminals.
6. If the effect of the KI on CSD has anything to do with glutamate clearance, and, specifically, the 50% reduction in the expression of GLT-1a, then one would expect that in the heterozygous null for GLT-1, one would see the same effect on CSD. While not essential for this paper, such an experiment would improve the paper because it would be a direct test of the hypothesis.
7. The idea that relatively small changes in GLT-1 expression have significant physiology effects suggests that there is no real "margin of safety" for the expression of GLT-1. GLT-1 represents 1% of brain protein, and one explanation for this high level of expression is that it is so important to prevent excitotoxicity that there is a huge margin of safety built in. Maybe this isn't true, and the brain really needs all the GLT-1 that it makes. This is an important point and if we can know the answer here it would go a long way to helping us understand how GLT-1 and glutamate clearance/glutamate binding functions in the brain
9. Ceftriaxone does many things. Does it affect alpha2 NKA expression? It does seem to affect expression of the glutamate-cystine antiporter xCT:
 Lewerenz J, Albrecht P, Tien ML, Henke N, Karumbayaram S, Kornblum HI, Wiedau-Pazos M, Schubert D, Maher P, Methner A (2009) Induction of Nrf2 and xCT are involved in the action of the neuroprotective antibiotic ceftriaxone in vitro. *J Neurochem* 111:332-343.
 Knackstedt LA, Melendez RI, Kalivas PW (2010) Ceftriaxone restores glutamate homeostasis and prevents relapse to cocaine seeking. *Biol Psychiatry* 67:81-84.
 Trantham-Davidson H, LaLumiere RT, Reissner KJ, Kalivas PW, Knackstedt LA (2012) Ceftriaxone normalizes nucleus accumbens synaptic transmission, glutamate transport, and export following cocaine self-administration and extinction training. *J Neurosci* 32:12406-12410.
 Rao PS, Sari Y (2014) Effects of ceftriaxone on chronic ethanol consumption: a potential role for xCT and GLT1 modulation of glutamate levels in male P rats. *J Mol Neurosci* 54:71-77.
 Alhaddad H, Das SC, Sari Y (2014) Effects of ceftriaxone on ethanol intake: a possible role for xCT and GLT-1 isoforms modulation of glutamate levels in P rats. *Psychopharmacology (Berl)* 231:4049-4057.
 Rao PS, Saternos H, Goodwani S, Sari Y (2015) Effects of ceftriaxone on GLT1 isoforms, xCT and associated signaling pathways in P rats exposed to ethanol. *Psychopharmacology (Berl)* 232:2333-2342.

Drugs have many effects. It is best to be skeptical about whether a drug's effect is due to its touted mechanism of action.

There are examples of papers that try rigorously to assess whether the action of ceftriaxone is due to an action on GLT-1, for example:

Melzer N, Meuth SG, Torres-Salazar D, Bittner S, Zozulya AL, Weidenfeller C, Kotsiari A, Stangel M, Fahlke C, Wiendl H (2008) A beta-lactam antibiotic dampens excitotoxic inflammatory CNS damage in a mouse model of multiple sclerosis. *PLoS one* 3:e3149.

Omrani A, Melone M, Bellesi M, Safiulina V, Aida T, Tanaka K, Cherubini E, Conti F (2009) Up-regulation of GLT-1 severely impairs LTD at mossy fibre--CA3 synapses. *The Journal of physiology* 587:4575-4588.

10. What happens to potassium clearance in the ceftriaxone treated animals?

11. The statement is made that the alpha2 NKA pumps exert a dampening role on CSD due to their role in glutamate clearance (and perhaps also their role in K⁺ clearance). But, the case really hasn't been made that the role in glutamate clearance is primary. The paper is really written as if the effect on glutamate clearance were primary. This conclusion has not been established or critically tested.

Referee #3 (Comments on Novelty/Model System):

The experiments are of high quality and for the most part the authors have used appropriate control experiments. What remains unclear is the cause-effect relationship in the last section of the paper.

Referee #3 (Remarks):

In this manuscript, the authors examine the role of astrocyte glutamate and K⁺-uptake in regulating cortical spreading depression in a mouse model of familial hemiplegic migraine type 2. The findings are intriguing, as the authors report reduced rate of glutamate uptake in KI mice, partial rescue of this effect with the antibiotic ceftriaxone. What remains puzzling to me is the last part, in which the authors suggest a reduced rate of K⁺-uptake in astrocytes of KI mice. The causal link that relates these findings is hard to follow, in the eyes of this reviewer.

Major points

The authors claim that the slowing of the STC following repetitive stimulations is more pronounced in FHM2 KI than in WT mice. When looking at the actual numbers, however, one would like to have additional tests of significance. In particular: is 1.18 +/- 0.03 (n=20) significantly different from 1.26 +/- 0.03 (n=21)? What is the p-value? My impression is that the rate of glutamate uptake is indeed slower in KI mice, but I am not entirely convinced that this difference is exacerbated after trains. I can be convinced with statistics.

There are some concerns about the accuracy with which the area of overlap between GLT-1a⁺ and VGLUT1⁺ puncta can be measured. The size of the region of overlap is prone to biases due to the 3D orientation of GLT-1a and VGLUT1 puncta and by the limited x-y and z-resolution of confocal microscopes. Small punch are likely missed and this could also bias the interpretation of the results. I would be hesitant to draw any strong conclusion for this data set.

ceftriaxone is an antibiotic and is therefore likely to exert other effects in addition to changing the expression of glial glutamate transporters. The manuscript currently lacks a clear indication that the greater severity of CSD is due to changes in glutamate uptake. Would it be possible to derive a dose-response curve for CSD threshold and propagation speed using different sub-saturating concentrations of TBOA?

It is a little surprising that the authors consider ceftriaxone "inefficient" at rescuing CSD spread (line 338). I would have said it is somewhat efficient since it can recover features of GLT1a expression and of CSD spread that are similar to the control group.

I agree that there is a significant K⁺-efflux from receptors, but the K⁺-current that accompanies STCs is due to K⁺-efflux during action potential generation. Therefore the authors should confirm that there is a significantly larger K⁺-current also in the presence of glutamate receptor blockers.

Minor points

Page 6. "[...] in layer 1 at 200 μm from" should be "[...] in layer 1, 200 μm from [...]"

Page 7. "These data indicate that glutamate released by an action potential (AP) at cortical synapses is taken up more slowly by astrocytes in FHM2 KI compared to WT mice." There is no indication that the stimulus intensity used by the authors only evoked one action potential. This statement should be removed.

Line 167. "a train of 10 APs". This is not really what the authors use: they use a train of 10 stimuli but do not know how many action potentials the train evokes.

The authors should express t_{10}/t_1 using the same units for WT and KI. Currently they are 1.18 and 40%.

The authors should cite the reference that provides a full description of the STC analysis (Scimemi and Diamond 2013, JoVe).

Line 206: "ir" should be "immuno-reactivity".

Page 10: "the density of gold particles coding for GLT-1a". Please rephrase: the gold particles do not code for GLT-1a.

Lines 241-242. "This correlates with, and likely reflects, 241 the absence of $\alpha 2$ NKA in cortical axon terminals (Cholet et al., 2002)." The expression of GLT-1a in axon terminals can also happen in the absence of $\alpha 2$ NKA. Please rephrase.

Lines 236-238: "The reduction of GLT-1a density in the membrane of PAPs mirrors the about 50% reduction of $\alpha 2$ NKA protein level in cortical crude synaptic membranes from W887R/+ KI mice revealed by Western blotting (Leo et al., 2011) (data not shown)." Please rephrase.

"For complete rescue one should have measured a 63% increase in CSD threshold in ceftriaxone-treated FHM2 KI mice, assuming that the facilitation of CSD induction in acute cortical slices from P30-33 saline-injected FHM2 KI mice is similar to that shown in slices from P34-35 KI mice (as suggested by the similar CSD threshold: 145 vs 149 ms)." Unclear; please rephrase.

Figure 6A: the y-axis label is "volume", but it is unclear whether this is really the case. I thought this would be the band intensity value in the Western blot experiments. Please check.

1st Revision - authors' response

29 March 2016

Point-by-point responses to the referees' comments

Referee n. 1

.....The work is well written and all studies appear to have followed sound methods and analytical approaches.

We thank the reviewer for the positive comments and the constructive criticisms that helped to improve the manuscript. Our detailed responses follow.

The manuscript in my opinion contains two major shortcomings. The authors overemphasize the observed effects of the FHM2 mutation on glutamate transporter function as a key mechanism underlying CSD susceptibility, while not stressing a role for the observed reduction in astrocytic K^+ conductance.

Major

1. The authors overemphasize their findings on the reduced astrocytic glutamate transporter current and the effects of ceftriaxone on CSD susceptibility.

We agree with the reviewer that in the original manuscript we overemphasized the role of the reduced rate of glutamate clearance by astrocytes in CSD facilitation in FHM2 KI mice. Indeed, the finding that, in contrast with the increased expression of GLT-1a in ceftriaxone-treated WT mice, neither the total expression of GLT-1a in the cortex nor the membrane density of GLT-1a in PAPs

were significantly increased in ceftriaxone-treated FHM2 KI mice (Figure 8 of revised manuscript, including new panel 8C), made it difficult to draw a clearcut conclusion regarding the role of impaired glutamate clearance by astrocytes in CSD facilitation. Following the suggestion of reviewer n. 3, in the revised manuscript we have used a second different approach to investigate whether there is a causative relationship between reduced rate of glutamate clearance and facilitation of CSD. We investigated whether pharmacological inhibition of a fraction of glutamate transporters in WT mice lowered the threshold for CSD induction and increased the velocity of CSD propagation and, having found that it did, we investigated whether pharmacological reduction of the rate of glutamate clearance in WT mice to a value similar to that of FHM2 KI mice reduced the threshold for CSD induction and increased the velocity of CSD propagation to values similar to those measured in the FHM2 mutants. The findings (new Figure 9; see Results p. 15 lines 357-392 and Discussion p. 20 lines 473-486) support the conclusion that the reduced rate of glutamate clearance by astrocytes can account for most of the facilitation of CSD initiation in FHM2 KI mice, leaving little room for other contributing mechanisms. In contrast, the reduced rate of glutamate clearance by astrocytes can account for only a fraction (although relatively large, around 60%) of the facilitation of CSD propagation in FHM2 KI mice, suggesting that other mechanisms contribute. It seems reasonable to expect that the reduced rate of K⁺ clearance by astrocytes in FHM2 KI mice would increase the rate of CSD propagation and thus contributes to the facilitation of CSD propagation. In contrast, according to our data, a contribution of the reduced rate of K⁺ clearance by astrocytes to the facilitation of CSD initiation in FHM2 mutants, if present, is small. As we discuss at page 21 (lines 504-512) this would be consistent with a more relevant role of the neuronal $\alpha 3$ NKA relative to the glial $\alpha 2$ NKA in K⁺ buffering during the CSD-inducing stimuli (high KCl pulses in our case). Evidence that this is the case during the CSD depolarization is provided by reports that the duration of the CSD depolarization is increased in heterozygous KI mice carrying a mutation that reduces the activity of the $\alpha 3$ NKA (Hunanyan et al, 2015), whereas, in contrast, the CSD duration is not affected in FHM2 KI mice with reduced $\alpha 2$ NKA (Leo et al, 2011).

1. Also, the calculation of the glutamate transporter current, which is one of the key readouts for FHM2 astrocytic function in this manuscript, needs to be better described.

We thank the reviewer for pointing out that the description of how we obtained the glutamate transporter current was not sufficiently clear. We have tried to clarify this in the revised manuscript as described in our response to point 1 iii).

i) Throughout the paper, the authors often refer directly to their experimental data as 'reduced glutamate clearance', while this is not what was actually measured. E.g. in the abstract line 27 'we show reduced rates of glutamate clearance' should state more precisely e.g. 'we showed reduced glutamate transporter currents, indicating reduced rates of glutamate clearance'.

Previous studies which analyzed the factors that shape the time course of the synaptically activated transporter current (STC) in hippocampal slices have shown that the time course of the STC reflects the time course of glutamate clearance (and hence the lifetime of the glutamate concentration profile at the astrocyte membrane) filtered by the electrotonic properties of astrocytes (Diamond 2005; Bergles and Jahr, 1997). Indeed, in hippocampal (Bergler and Jahr, 1997; Diamond, 2005; Diamond and Jahr, 2000) as well as cortical slices (cf Supplementary Figure 1 added to the revised manuscript), when the density of glutamate transporters is pharmacologically reduced, the decay kinetics of the STC becomes slower, as expected for a reduced rate of glutamate clearance and a longer lifetime of synaptically released glutamate in the extracellular space. Moreover, in correlation with the increased expression of glutamate transporters with increasing age of the animal, the decay kinetics of the STC becomes faster with increasing age, reflecting the increased rate of glutamate clearance due to the increased transport capacity (whereas transporter kinetics do not vary with age) (Bergles and Jahr, 1997; Diamond, 2005; Thomas et al, 2011).

Thus, the decay kinetics of the STC (and the time constant of the exponential function that best fits the decay, as measured in our study) does provide a measure of the rate of clearance of synaptically released glutamate. Due to the astrocyte electrotonic filtering, the rate of glutamate clearance obtained from the time constant of the exponential function that best fits the STC decay is an underestimate of the actual rate of clearance (Diamond, 2005), but for our scope this is not relevant because we are interested in relative differences in the rate of glutamate clearance in WT and FHM2 KI mice (not in the real values). Moreover, it is important to note that the decay kinetics of the STC provides a measure of the rate of clearance of synaptically released glutamate that is independent of the amount of glutamate released. In fact, in both hippocampal (Diamond and Jahr, 2000; Diamond, 2005) and cortical slices (Unichenko et al, 2012) (cf Supplementary Figure 1 added to the revised manuscript), changing the intensity of extracellular stimulation (and hence the number of stimulated fibers and the amount of glutamate released) changed the STC amplitude without affecting the STC decay kinetics.

We have revised the first part of the Results section (p. 6, lines 135-147) and added Supplementary Figure 1 to better explain why the decay kinetics of the STC provides a measure of the rate of clearance of synaptically released glutamate.

It would have been nice to have direct evidence for enhanced glutamate and K⁺ levels e.g. via glutamate or pH sensor measurements.

We agree that it would be nice to measure glutamate and K⁺ levels with specific sensors (ideally in vivo) and we plan to do this in the future, but at the moment, unfortunately, we are not set-up for this.

ii) The conclusions in the manuscript should be rephrased to be in line with the experimental data, that a) did not show an effect of ceftriaxone on astrocytic GLT-1 expression, b) show only a minor increase in CSD threshold upon ceftriaxone, without a change in CSD velocity, and c) indicated a

reduced rate of K⁺ clearance. For example, the conclusion of the Abstract the statement 'we obtain evidence that defective glutamate clearance contributes to facilitation of CSD initiation in FHM2,..' needs rephrasing in this respect, as well as the conclusions in the Discussion.

Please, see answer to major point 1. The manuscript has been revised to include the conclusions derived from the new experiments discussed above.

iii) The calculation of the synaptically-activated transporter current STC that is used throughout the paper and depicted first in Figure 1A and 1B, needs to be described better in the Methods and Results section and Figure legend. In the Methods on p.25, line 615, it is not clear which data are pooled. Were the 16 experiments from 16 slices of WT mice, or a pool of WT and KI recordings? To aid the understanding for the non-specialized readership, I suggest to add a schematic of the recording paradigm, indicating the site of stimulation and recording of the synaptically-activated glutamate transporter current STC. Such a schematic will also make clear that the STC data are interpreted as measure of astrocytic glutamate clearance.

We thank the reviewer for suggesting to add a schematic of the recording paradigm to help clarifying how the STC was obtained and its meaning. This schematic has been included in Figure 1A of the revised manuscript. To better describe how the STC was isolated, we have revised the text: both the Results section (p. 6, lines 128-133) and the Methods section (p. 26, lines 635-647), where we have also specified that the average normalized TBOA-insensitive current was obtained by pooling WT and KI recordings because the time course of the residual current in WT and KI slices was similar. As explained above, the first part of the Results section has been revised to better explain why the decay kinetics of the STC provides a measure of the rate of glutamate clearance. Moreover, to make clear how we obtained the time constant of decay of the STC, we have colored in red (rather than grey, which was probably barely visible) the exponential function that best fits the decay of the representative STC traces in Figures 1 and 2 (and have indicated the corresponding τ_{decay}), and have colored in green the exponential waveforms approximating the average TBOA-insensitive currents (and changed accordingly the Figure legends).

Further, effects of the FHM2 mutation on astrocytic Na⁺ homeostasis is not discussed, while this is not unlikely to influence CSD susceptibility as well.

2. The effect of the glial $\alpha 2$ Na⁺/K⁺ ATPase in reducing the intra-astrocytic sodium concentration (by Na⁺ efflux coupled to K⁺ influx) is not addressed, while it could also contribute to the enhanced CSD susceptibility in FHM2 mice. A predicted reduction in Na⁺/K⁺ ATPase function in FHM2 KI mice would be expected to influence astrocyte transporter mechanisms relying on Na⁺ influx. For example, apart from Na⁺ dependent glutamate uptake by astrocytes, glial Na⁺ influx is directly coupled to H⁺ extrusion and HCO₃⁻ uptake, via NHE and NBC transporters. Reduced activity of these transporters that may be expected with reduction in Na⁺/K⁺ ATPase

function in FHM2 KI mice could alter cortical pH which, in case of a more alkaline environment, would enhance neuronal excitability and can influence CSD susceptibility. These aspects need be addressed in the discussion on the CSD susceptibility experiments in cortical slices from FHM2 mice.

We thank the reviewer for pointing out that we did not discuss possible effects of the FHM2 mutation on astrocytic Na⁺ homeostasis and on astrocyte transporter mechanisms relying on Na⁺ influx, and how the possible effects on some of these transporters could contribute to the facilitation of CSD. We have now discussed this at page 22 (lines 528-538) and in the Appendix Supplementary Discussion (lines 50-70). We had to move part of this discussion to the appendix to be able to fit the text within an acceptable number of characters (since, due to the new added data, the revised manuscript much exceeded the maximal allowed number of characters).

Minor

a) The authors have not specified when n-numbers refer to number of slices or number of mice. I suggest to uniformly depict by 'n' the number of slices and indicate with 'N' the number of mice that were used for those slices.

As suggested by the reviewer, in the legends of the Figures of the revised manuscript we have indicated with “n” the number of slices and with “N” the number of mice from which the slices were obtained.

b) Ages of mice should be mentioned for each of the subfigures, e.g. age is not mentioned for the experiments from Figure 1 and Figure 6A

In the legends of the Figures of the revised manuscript we have mentioned the ages of the mice for each subfigure.

c) Methods: it is not clearly described whether electrophysiological studies and analyses were performed by a blinded observer, as was mentioned for the immunocytochemical studies.

In general we don't perform blind the electrophysiological experiments in brain slices, since they involve measurements of cellular currents with minimal possibility of bias of the results by the experimenter. Also the analysis of the electrophysiological experiments (mainly consisting in best fitting current decays with an exponential function in this study) is much less subject to possible bias than that of the immunocytochemical studies and usually is not performed blind. Nonetheless, in this study, a number of STC analyses were performed blind. Moreover, as specified in Appendix Supplementary Methods (lines 118-119), all CSD experiments and analyses in saline-treated and cef-treated mice and most of the CSD experiments in WT and KI mice were performed blind.

c) Results: the authors should give information on the astrocytic input resistance for WT compared to KI recordings, and mention # exclusions (as indicated in the Methods) based on variable or too high access resistance, to show that WT and KI astrocytes had comparable access resistance.

The WT and KI astrocytes from which we recorded from had comparable access resistances: $R_a = 14.7 \pm 0.6 \text{ M}\Omega$ for WT ($n = 66$) and $R_a = 16.0 \pm 0.8 \text{ M}\Omega$ for KI ($n = 47$), $p = 0.15$. Although we did not keep the record of the number of exclusions, we did not observe noticeable differences. Also input resistance was similar in WT and KI astrocytes (in the fraction of cells where we measured it): $R_i = 13.2 \pm 0.7 \text{ M}\Omega$ for WT ($n = 42$) and $R_i = 14.2 \pm 1.2 \text{ M}\Omega$ for KI ($n = 24$).

These informations have been introduced in the revised manuscript (at page 24, lines 605-606 and p. 25 lines 627-628).

d) Results Figure 5A: the delay from KCl puff to CSD appears longer for FHM1 KI compared to WT mice. This parameters needs to be mentioned in the

The delay from KCl puff to CSD is somewhat variable in both WT and FHM2 KI mice since it mainly depends on how close to CSD threshold is the KCl puff: the more suprathreshold is the puff the smaller is the delay. Thus, the delay is relatively insensitive to the rate of CSD propagation. The choice of representative WT IOS traces in the original Figure 5A was not the best, since, as the reviewer noticed, the delay in the WT traces was particularly short (and in this respect not representative). The equivalent Figure 6A of the revised manuscript shows WT IOS traces (from a different experiment) that are more representative and show a delay similar to that in FHM2 KI traces.

d) Discussion: given the described neuronal expression of Na^+/K^+ ATPase during embryonic development, possible effects of the FHM2 mutation on cortical synaptic function in adulthood, e.g. as a result of alterations in cortical network development, should be addressed.

We have addressed this possibility in the revised manuscript in the Appendix Supplementary Discussion (lines 63-67).

Referee n.2

This is a very interesting, carefully executed, well-written and potentially important paper. It focuses onThe recognition that GLT-1a is expressed in axon terminals, and the demonstration that its expression in this location is not changed in the KI, as expected given that $\alpha 2$ subunit is astrocytic and not neuronal, is very nice. Also, the authors have shown, probably for the first time, that ceftriaxone increases GLT-1a protein expression in axon terminals.

We thank the reviewer for the positive comments and the constructive criticisms that helped to improve the manuscript. Our detailed responses follow.

The central issue here is whether facilitation of CSD is due to glutamate clearance abnormality or the K clearance abnormality-or some other abnormality, an issue that remains unresolved.

We agree with the reviewer that the issue of whether facilitation of CSD is due to glutamate clearance abnormality or the K clearance abnormality, remained unresolved in the original manuscript. In particular, the finding that, in contrast with the increased expression of GLT-1a in ceftriaxone-treated WT mice, neither the total expression of GLT-1a in the cortex nor the membrane density of GLT-1a in PAPs were significantly increased in ceftriaxone-treated FHM2 KI mice (Figure 8 of revised manuscript, including new panel 8C), made it difficult to draw a clearcut conclusion regarding the role of impaired glutamate clearance by astrocytes in CSD facilitation. Following the suggestion of reviewer n. 3, in this revised paper we have used a second different approach to investigate whether there is a causative relationship between reduced rate of glutamate clearance and facilitation of CSD. We investigated whether pharmacological inhibition of a fraction of glutamate transporters in WT mice lowered the threshold for CSD induction and increased the velocity of CSD propagation and, having found that it did, we investigated whether pharmacological reduction of the rate of glutamate clearance in WT mice to a value similar to that of FHM2 KI mice reduced the threshold for CSD induction and increased the velocity of CSD propagation to values similar to those measured in the FHM2 mutants. The findings (new Figure 9; see Results p. 15 lines 357-392 and Discussion p. 20 lines 473-486) support the conclusion that the reduced rate of glutamate clearance by astrocytes can account for most of the facilitation of CSD initiation in FHM2 KI mice, leaving little room for other contributing mechanisms. In contrast, the reduced rate of glutamate clearance by astrocytes can account for only a fraction (although relatively large, around 60%) of the facilitation of CSD propagation in FHM2 KI mice, suggesting that other mechanisms contribute. It seems reasonable to expect that the reduced rate of K⁺ clearance by astrocytes in FHM2 KI mice would increase the rate of CSD propagation and thus contribute to the facilitation of CSD propagation. Our data suggest that a contribution of the reduced rate of K⁺ clearance by astrocytes to the facilitation of CSD initiation in FHM2, if present, is small.

There are some problems that need to be addressed:

1. A primary observation is that there is a 20% reduction in the mean size of GLT-1a + puncta in the FHM2 KI mice. Does this mean that the size of spines is smaller? It is not clear that a decrease in protein expression of GLT-1a would make for a decrease in size of the structures that express GLT-1a. A change in spine size might be important for the pathophysiology of CSD.

In the last decade, several studies re-analyzed GLT-1 localization at excitatory synapses (Chen et al 2004; Holmseth et al, 2009; Melone et el, 2009). These studies showed that GLT-1a is expressed in

astroglial cells and neurons and that, at excitatory synapses, it is confined at perisynaptic astrocytic processes and axon terminals. None of the studies reported GLT-1 at spines. We believe that the reviewer meant perisynaptic astrocytic processes and not spines, and we discuss his point based on this interpretation.

The problem raised by the referee, i.e., whether the size of an immunopositive puncta (which are typically visualized by light confocal microscopy) reflects the size of a specific subcellular compartment or the amount of a given immunodetected protein, is central to much of our analysis. In most studies (e.g., Bozdagy et al, 2000; Bragina et al, 2006; Bellesi et al., 2009; Omrani et al., 2009; Antonova et al, 2001), the increase of size of positive puncta has been considered strongly suggestive of an increase of the expression of the protein studied and not of the dimension of subcellular compartment expressing that protein. Nevertheless, prompted by the reviewer comment, we sought to exclude that 20% reduction in the mean size of GLT-1a + puncta in the FHM2 KI mice reflects a reduction of the size of perisynaptic astrocytic processes (PAP) and/or axon terminals (AxT) expressing GLT-1. For this purpose, we analyzed and compared the area of PAP and AxT of all sampled profiles used for immunogold analysis of GLT-1 (Table 1), and found that mean areas of PAP and AxT in WT and FHM2 KI were comparable (PAP: 0.33 ± 0.09 vs 0.38 ± 0.05 ; $P=0.55$; AxT: 0.71 ± 0.17 vs 0.56 ± 0.10 ; $P=0.26$). This has been included in the Legend of Table 1 of the revised manuscript. This observation supports the interpretation of Bozdagy (2000), Antonova (2001) and of many others, and is strongly suggestive that the reduction of GLT-1 puncta is associated to a reduction of GLT-1 levels.

2. The authors observe that with increasing temperature, the threshold for CSD induction decreases. Since the activity of glutamate transporters has a Q10 of 2-3, these data re temperature effect seem hard to rationalize. Perhaps the writing is just confusing; compare what is stated on p. 12 with the last sentence of same paragraph.

The threshold for CSD induction decreases (29%) with increasing temperature (from room T to 30 °C) in P34-35 FHM2 KI mice, while in WT mice the CSD threshold is much less sensitive to the change in temperature (it decreases only 9%). We rationalize this finding as follows. With increasing temperature many physiological processes accelerate including glutamate release, metabolism etc and the high Q10 of glutamate transporters and also of Na, K ATPases ensure that glutamate uptake and active transport by the Na, K ATPases keep up with the physiological changes in WT mice (and hence the CSD threshold is not much affected). As a consequence of the reduced expression of the Na, K ATPases and of the glutamate transporters, this occurs less efficiently in FHM2 KI mice and results in a lower threshold for CSD induction with increasing temperature in the mutants; therefore, the facilitation of CSD induction in KI relative to WT increases with increasing temperature.

Since the data on the temperature and age dependence of the CSD facilitation in FHM2 KI mice are not essential to follow the main thread of the paper, and given that with the new data added to the

paper we much exceeded the character limit of the manuscript, we have moved the data in Figure 5D and 5C of the original manuscript into Supplementary Figure 2 of the revised manuscript.

3. The authors observed, with ceftriaxone, a 50% increase in protein on Western blots. Some labs have not observed an effect of cef on normal animals. These data need to be normalized to an internal standard.

We performed new Western blots experiments in WT mice using an internal standard, and in our experimental conditions we confirmed the increase of GLT-1 in the cortex of C57BL/6 mice following ceftriaxone treatment (see Figure 7A of revised manuscript), in line with our previous data and with numerous published studies in both mice and rats. Moreover, we also performed Western blots in FHM2 KI mice and these new experiments show that, in contrast with WT, GLT-1 level does not significantly increase following ceftriaxone-treatment in FHM2 KI mice (Figure 8C of revised manuscript).

We would like to thank the reviewer for his comment that gave us the opportunity to verify our data regarding the ceftriaxone effect on GLT-1 using housekeeping proteins as internal standard.

4. Cef seems to increase the size of puncta without increasing the density of GLT-1a gold particles. This is hard to understand in view of: 1) the increase on western blot; 2) the change in size of GLT-1a puncta, if, indeed, the size of GLT-1a puncta reflect anything about GLT-1a expression, and not a change in size of something else, like spine size.

Consistently with previous studies (Bellesi et al., 2008; Omrani et al., 2009), Cef increases GLT-1 protein (Figure 7A) and size of GLT-1/VGLUT1 related puncta (Figure 7B) in WT mice. The findings in Fig 8 show that, in contrast, in FHM2 KI mice, Cef does not increase GLT-1 protein (new WB data, Fig 8C), does not modify size of GLT-1/VGLUT1 related puncta (Fig. 8D) and does not modify immunogold density of GLT-1 in the membrane of PAP (Fig 8A), but it does increase GLT-1 density in the membrane of axon terminals, AxTs (Fig. 8B). In checking the statistics before submission of the revised manuscript we realized that an inappropriate test was used for analyzing the data in Table I and II; application of the appropriate Mann-Whitney U test made significantly different also the increase in total density (besides membrane density) of GLT-1a in AxTs in Cef-treated vs saline-treated FHM2 KI mice (Fig 8B) (and also made significant the peculiar 10% decrease of GLT-1a density in the cytoplasm of PAPs of Cef-treated FHM2 KI mice, a difficult to interpret result which does not affect our main conclusions; Fig 8A). As written at page 14 (lines 338-342), given the relatively small fraction of GLT-1 at AxTs relative to that in PAPs (and the limit of resolution of confocal microscopy) the unaltered GLT-1 protein and unaltered size of overlapping GLT-1a/VGLUT1 puncta are consistent with the increase of GLT1a only at AxTs.

We realize that, probably, the labeling of Figures 6 and 7 of the original manuscript (which have become Figs 7 and 8 of the revised manuscript) was misleading because we used the same abbreviation (Ctr) to indicate in the original Figures 6A and 6B the saline-treated WT mice and in

Figures 6C and 7 the saline-treated FHM2 KI mice. In Figures 7 and 8 of the revised manuscript we have changed the labels with Ctr WT in Figure 7A,B and Ctr KI in Figures 7C and 8.

5. The authors suggest that the partial rescue of the facilitation of CSD in cef treated animals is likely due to increased reuptake in terminals, because there is a change in density of GLT-1a immunogold in terminals but not in puncta. But, this is hard to believe, given that only 5-10% of GLT-1a is expressed in terminals.

The Discussion of the revised manuscript has been changed (p. 19 lines 462-471) to take into account this comment (as well as the second part of comment 9) of the reviewer. See also our response to comment 9.

6. If the effect of the KI on CSD has anything to do with glutamate clearance, and, specifically, the 50% reduction in the expression of GLT-1a, then one would expect that in the heterozygous null for GLT-1, one would see the same effect on CSD. While not essential for this paper, such an experiment would improve the paper because it would be a direct test of the hypothesis.

We agree with the reviewer that this would be an interesting and informative experiment and, in fact, it is in our plans to perform it in the future, thanks to a collaboration that we have recently established to obtain heterozygous GLT-1 null mice. Unfortunately, we don't have this mouse available in our labs at the moment, and obtaining it and being able to use it for these experiments would have required a considerable amount of time (beyond that allowed for a revision). However, as mentioned above, we measured the effect on CSD of pharmacological inhibition of a fraction of glutamate transporters producing a slowing of the rate of glutamate clearance in WT mice similar to that in FHM2 KI mice, and have included in the revised manuscripts the results of these experiments, which support the conclusion that the reduced rate of glutamate clearance by astrocytes can account for most of the facilitation of CSD initiation in FHM2 KI mice.

7. The idea that relatively small changes in GLT-1 expression have significant physiology effects suggests that there is no real "margin of safety" for the expression of GLT-1. GLT-1 represents 1% of brain protein, and one explanation for this high level of expression is that it is so important to prevent excitotoxicity that there is a huge margin of safety built in. Maybe this isn't true, and the brain really needs all the GLT-1 that it makes. This is an important point and if we can know the answer here it would go a long way to helping us understand how GLT-1 and glutamate clearance/glutamate binding functions in the brain

We agree with the reviewer that this is an important point. Combining together the reduced GLT-1 expression in perisynaptic astrocytic processes, the reduced rate of glutamate clearance by astrocytes and the reduced threshold of CSD induction in FHM2 KI mice with the reduced threshold of CSD induction in WT mice after pharmacological inhibition of a fraction of glutamate transporters, our

data support the conclusion that around 50% reduction of GLT-1 expression in the astrocyte processes surrounding glutamatergic synapses does lower the threshold for CSD induction. Nonetheless, we think that our data also suggest that actually there is “a margin of safety” for the expression of GLT-1 (although perhaps not so huge), because a 50% reduction of GLT-1 expression in perisynaptic processes leads to only a 21% reduction of the rate of glutamate clearance by astrocytes following a single pulse stimulation.

9. Ceftriaxone does many things. Does it affect alpha2 NKA expression? It does seem to affect expression of the glutamate-cystine antiporter xCT:

We performed new WB experiments in FHM2 KI mice following ceftriaxone treatment to investigate whether ceftriaxone affects $\alpha 2$ NKA and xCT expression. The results show that in the KI mice, ceftriaxone has no effects on $\alpha 2$ NKA and xCT (Expanded View Figure 2; Results p. 15 lines 346-355).

In the papers cited by the reviewer, GLT-1 and xCT appears to be co-regulated in nucleus accumbens in rats with a history of cocaine self-administration or ethanol consumption; the expression of both GLT-1 and xCT is decreased after chronic cocaine or ethanol consumption (and also after extinction from cocaine), and ceftriaxone restores the expression and function of both GLT-1 and xCT. However, the co-regulation of GLT-1 and xCT is not a general finding, since Shen et al (2014, J Neurosci) reported that in rats with a history of heroin self-administration the decrease of GLT-1 expression in nucleus accumbens after extinction training is not accompanied by a decrease of XCT expression (that actually appears to increase), and ceftriaxone treatment rescues Na-dependent glutamate uptake without affecting Na-independent glutamate uptake (indicating no change in xCT function). In our FHM2 KI mice, neither GLT-1 nor xCT levels in cerebral cortex are affected by ceftriaxone treatment.

Drugs have many effects. It is best to be skeptical about whether a drug's effect is due to its touted mechanism of action.

We agree with the reviewer and in the revised manuscript we have discussed the possibility that some other effect of ceftriaxone, not related to GLT-1 expression, might underlie its small effect on CSD threshold in FHM2 KI mice. This could provide an explanation for the small CSD rescue effect of ceftriaxone which is alternative to that of the increased reuptake of glutamate in axon terminals (that seems unlikely in view of the small fraction of GLT-1 expressed in terminals) (Discussion p. 19 lines 463-472). However, the parallelism between the differential effect of ceftriaxone on CSD threshold and CSD velocity in FHM2 KI mice (small rescue of threshold no effect on velocity) and the differential effect of inhibition of a fraction of glutamate transporters on CSD threshold and velocity in WT mice (suggesting that the reduced rate of glutamate clearance can account for most of the reduced CSD threshold, but only a fraction of the increased CSD velocity in FHM2 KI mice) appears consistent with the interpretation that the increased reuptake of glutamate in axon terminals

might account for the small rescue of facilitation of CSD induction produced by ceftriaxone (Discussion p. 20, lines 477-486).

There are examples of papers that try rigorously to assess whether the action of ceftriaxone is due to an action on GLT-1, for example:

Melzer N, Meuth SG, Torres-Salazar D, Bittner S, Zozulya AL, Weidenfeller C, Kotsiari A, Stangel M, Fahlke C, Wiendl H (2008) A beta-lactam antibiotic dampens excitotoxic inflammatory CNS damage in a mouse model of multiple sclerosis. PloS one 3:e3149.

Omrani A, Melone M, Bellesi M, Safiulina V, Aida T, Tanaka K, Cherubini E, Conti F (2009) Up-regulation of GLT-1 severely impairs LTD at mossy fibre--CA3 synapses. The Journal of physiology 587:4575-4588.

The papers mentioned by the reviewer used DHK to assess whether the ceftriaxone-induced attenuation of the clinical symptoms in a model of EAE (Melzer et al) or the ceftriaxone-induced impairment mGluR-dependent LTD (Ormani et al) were due to ceftriaxone-induced upregulation of GLT-1. Unfortunately, this approach does not seem feasible in our case to establish whether the ceftriaxone-induced small rescue of CSD in FHM2 KI mice is due to upregulation of GLT-1 because, given our results with subsaturating concentrations of TBOA in WT mice, we expect that DHK per se would much decrease the CSD threshold in FHM2 KI mice.

10. What happens to potassium clearance in the ceftriaxone treated animals?

The animals after ceftriaxone treatment have at least 30-33 days of age, and at this age it is very difficult to obtain acute cortical slices of sufficient quality to perform good recordings from astrocytes that allow reliable measurements of the time constant of decay of IK. Therefore, for technical reasons, we were unable to measure the effect of ceftriaxone on potassium clearance. However, we performed Western blot experiments to measure the expression of Kir4.1 in FHM2 KI mice before and after ceftriaxone treatment. According to the literature, Kir4.1 is the glial K⁺ channel that accounts for most of IK measured in voltage-clamped astrocytes and plays a key role in K⁺ spatial buffering. The results show that ceftriaxone has no effect on the protein level of Kir4.1 in the cerebral cortex of FHM2 KI mice.

11. The statement is made that the alpha2 NKA pumps exert a dampening role on CSD due to their role in glutamate clearance (and perhaps also their role in K⁺ clearance). But, the case really hasn't been made that the role in glutamate clearance is primary. The paper is really written as if the effect on glutamate clearance were primary. This conclusion has not been established or critically tested.

Please, see response to the initial general comment of the reviewer.

Referee n. 3

The experiments are of high quality and for the most part the authors have used appropriate control experiments.

We thank the reviewer for the positive comments and the constructive criticisms that helped to improve the manuscript. Our detailed responses follow.

What remains unclear is the cause-effect relationship in the last section of the paper.

Referee #3 (Remarks):

In this manuscript, the authors examine the role of astrocyte glutamate and K⁺-uptake in regulating cortical spreading depression in a mouse model of familial hemiplegic migraine type 2. The findings are intriguing, as the authors report reduced rate of glutamate uptake in KI mice, partial rescue of this effect with the antibiotic ceftriaxone. What remains puzzling to me is the last part, in which the authors suggest a reduced rate of K⁺-uptake in astrocytes of KI mice. The causal link that relates these findings is hard to follow, in the eyes of this reviewer.

We are not sure we understand this comment of the reviewer. In particular, we are not sure whether the “causal link that relates these findings” refer to the findings of a reduced rate of glutamate uptake and a reduced rate of K⁺ uptake in astrocytes of FHM2 KI mice or refer to the reduced rate of K⁺ uptake and the facilitation of CSD in FHM2 KI mice.

In the first case, there is no causal link between reduced rate of glutamate uptake and reduced rate of K⁺ uptake; but both alterations are caused (independently) by the reduced expression of the $\alpha 2$ Na, K-ATPase produced by the FHM2 mutation. The reduced expression of the $\alpha 2$ Na, K-ATPase results in a reduced expression of GLT-1a and a reduced rate of glutamate uptake into astrocytes. Given the active pumping of K⁺ ions inside astrocytes mediated by the $\alpha 2$ Na, K-ATPase, its reduced expression also results in reduced rate of clearance of extracellular K⁺ (thus showing that the $\alpha 2$ Na, K-ATPase contributes to K⁺ buffering during neuronal activity).

In the second case, considering what is known regarding the mechanisms of initiation and propagation of CSD, in general a reduced rate of K⁺ clearance is expected to contribute to CSD facilitation. However, the extent to which a reduced rate of K⁺ uptake in astrocytes due to reduced expression of the $\alpha 2$ Na, K-ATPase contributes to lower the CSD threshold is expected to depend on the role that the glial $\alpha 2$ pumps play in K⁺ buffering during the CSD inducing stimuli relative to the role played by the neuronal $\alpha 3$ Na, K-ATPases and by other mechanisms contributing to K⁺ clearance. The new findings obtained using subsaturating concentrations of TBOA in WT mice (following the suggestion of this reviewer; see our response to the corresponding comment below and also response to major point 1 of referee n. 1) support the conclusion that the reduced rate of glutamate clearance by astrocytes can account for most of the facilitation of CSD initiation in FHM2 KI mice, leaving little room for a contribution of the reduced rate of K⁺ uptake by astrocytes in facilitation of CSD initiation. In contrast, the reduced rate of glutamate clearance can account for

only a fraction (although relatively large) of the facilitation of CSD propagation, and it seems reasonable to expect that the reduced rate of K⁺ clearance by astrocytes in FHM2 KI mice would increase the rate of CSD propagation and thus contribute to the facilitation of CSD propagation (see Discussion lines p. 20 lines 473-486, p. 21 lines 504-512, 524-526).

At any rate, prompted by this comment of the reviewer, we realized that the Results section flows better if we, first, present the analysis of the functional consequences of the FHM2 mutation on both glutamate and K⁺ clearance by astrocytes, and then present the approaches aimed at investigating whether there is a cause-effect relationship between reduced rate of glutamate clearance and facilitation of CSD. Hence, in the revised manuscript the original last Fig. 8 (comparing the decay kinetics of IK in WT and KI) has become Fig 5 of the revised manuscript (and the original Figs 5, 6, 7 have become Figs 6, 7, 8).

Major points

The authors claim that the slowing of the STC following repetitive stimulations is more pronounced in FHM2 KI than in WT mice. When looking at the actual numbers, however, one would like to have additional tests of significance. In particular: is 1.18 ± 0.03 (n=20) significantly different from 1.26 ± 0.03 (n=21)? What is the p-value? My impression is that the rate of glutamate uptake is indeed slower in KI mice, but I am not entirely convinced that this difference is exacerbated after trains. I can be convinced with statistics.

We thank the reviewer for pointing out that in the original manuscript we did not include the p value of the test of significance of the difference between the τ_{10}/τ_1 (50 Hz) values measured in WT and KI mice: 1.18 ± 0.03 (n=20) and 1.26 ± 0.03 (n=21). According to the unpaired t-test the two values are significantly different with p=0.036. We have now measured τ_{10}/τ_1 (50 Hz) in 3 new experiments in WT mice; their inclusion in the revised manuscript did not change the average value of τ_{10}/τ_1 (50 Hz) but reduced the SEM (1.18 ± 0.02 , n=23) thus improving the p value (p= 0.021). We have performed 7 new experiments using 100 Hz trains in WT slices. The τ_{10}/τ_1 (100 Hz) values measured in WT and KI mice are 1.25 ± 0.03 , n=18, and 1.40 ± 0.05 , n=14, respectively; these values are significantly different with p= 0.009. In the revised Results section (p. 8 lines 168-184) we have reported these values and have tried to describe more clearly our findings (to address one of the minor points of the reviewer).

There are some concerns about the accuracy with which the area of overlap between GLT-1a+ and VGLUT1+ puncta can be measured. The size of the region of overlap is prone to biases due to the 3D orientation of GLT-1a and VGLUT1 puncta and by the limited x-y and z-resolution of confocal microscopes. Small punch are likely missed and this could also bias the interpretation of the results. I would be hesitant to draw any strong conclusion for this data set.

In the double immunofluorescence experiments we have measured the size of GLT-1a puncta (green in Figures 6 and 7) that overlap with the VGLUT1 puncta, not the size of the area of overlap between the puncta (yellow in Figures 6 and 7). Anyway, needless to say, we agree with the reviewer that these data cannot be used to draw strong conclusions, although they are consistent with the post-embedding EM data, on which we base our strong conclusions regarding the membrane expression of GLT-1a.

Ceftriaxone is an antibiotic and is therefore likely to exert other effects in addition to changing the expression of glutamate transporters. The manuscript currently lacks a clear indication that the greater severity of CSD is due to changes in glutamate uptake. Would it be possible to derive a dose-response curve for CSD threshold and propagation speed using different sub-saturating concentrations of TBOA?

We agree with the reviewer that the original manuscript lacked a clear indication that the facilitation of CSD in the FHM2 KI mice is due to the changes in glutamate uptake, and thank very much the reviewer for suggesting the use of subsaturating concentrations of TBOA to inhibit a fraction of glutamate transporters in wild-type mice as a second different approach to investigate whether there is a causative relationship between reduced rate of glutamate clearance and facilitation of CSD. Following this suggestion, we first identified a concentration of TBOA that increased by 32 % the time constant of decay of the STC in WT astrocytes, thus producing a slowing of the rate of glutamate clearance close to (although larger than) that produced by the FHM2 mutation, and tested its effect on CSD threshold and velocity in WT mice. Having found that this TBOA concentration produced a facilitation of CSD induction in WT mice larger than that produced by the FHM2 mutation in KI mice (and a facilitation of CSD propagation close to that in FHM2 KI mice), we then identified the concentration of TBOA that produced a slowing of glutamate clearance quantitatively similar to that produced by the FHM2 mutation and measured its effect on CSD threshold and velocity. The findings (new Figure 9; see Results p. 15 lines 356-392 and Discussion p. 20 lines 473-486) support the conclusion that the reduced rate of glutamate clearance by astrocytes can account for most of the facilitation of CSD initiation in FHM2 KI mice, leaving little room for other contributing mechanisms. In contrast, the reduced rate of glutamate clearance by astrocytes can account for only a fraction (although relatively large, around 60%) of the facilitation of CSD propagation in FHM2 KI mice, suggesting that other mechanisms contribute.

It is a little surprising that the authors consider ceftriaxone “inefficient” at rescuing CSD spread (line 338). I would have said it is somewhat efficient since it can recover features of GLT1a expression and of CSD spread that are similar to the control group.

If ceftriaxone was efficient at rescuing CSD spread one would have expected a decrease of the velocity of CSD propagation in ceftriaxone-treated FHM2 KI mice (to values similar to those in WT mice for a complete rescue). In contrast, Figure 7C of the revised manuscript shows that ceftriaxone

does not affect the rate of CSD propagation in FHM2 KI mice, since the velocity is almost identical in ceftriaxone-treated and saline-treated FHM2 KI mice (which are the controls here). We realize that, probably, the labeling of Figure 6 of the original manuscript was misleading because we used the same abbreviation (Ctr) to indicate in Figure 6A and 6B the saline-treated WT mice and in Figure 6C the saline treated FHM2 KI mice. In Figures 7 of the revised manuscript we have changed the labels with Ctr WT in Figure 7A,B and Ctr KI in Figure 7C.

Similarly, if ceftriaxone was efficient at increasing the expression of GLT-1a in FHM2 KI mice (as it is in WT mice, as shown in Figure 7A,B), then one would have expected an increase of the GLT-1a protein expression and of the size of the GLT-1a+ puncta that overlap with VGLUT1 in the cerebral cortex of FHM2 KI mice, similar to the increase observed in WT mice. In contrast, Figure 8C (new) and Figure 8D (that was 7C in the previous Fig 7) show that neither the GLT-1a protein expression nor the size of the GLT-1a+ puncta that overlap with VGLUT1 are significantly increased in ceftriaxone-treated relative to saline-treated FHM2 KI mice. To avoid confusion, we have relabeled also the original Figure 7 (which is Fig 8 in the revised manuscript).

I agree that there is a significant K-efflux from receptors, but the K-current that accompanies STCs is due to K-efflux during action potential generation. Therefore the authors should confirm that there is a significantly larger K-current also in the presence of glutamate receptor blockers.

We assume that here the reviewer means that we should confirm that the time constant of decay of the K current is significantly larger in FHM2 KI mice also in the presence of glutamate receptor blockers, since this is what we have shown in the absence of glutamate receptor blockers in Figure 8 of the original manuscript (Fig 5 of revised manuscript). Our conclusion that the reduced membrane expression of the $\alpha 2$ Na,K-ATPase in the FHM2 KI mice causes a reduced rate of K⁺ clearance by cortical astrocytes is based on the slower decay of I_K following a 50 Hz train of pulses in the absence of glutamate receptor blockers (since it has been previously shown that the time constant of decay of I_K provides a measure of the rate of K⁺ clearance that is equivalent to that obtained with [K]_e-sensitive microelectrodes: Meeks and Mennerick, 2007). The analysis of the decay of I_K following a 50 Hz train of pulses in the presence of glutamate receptor blockers ($I_K \tau_{\text{decay}} = 2.38 \pm 0.09$ s, n=18, in WT vs $I_K \tau_{\text{decay}} = 2.95 \pm 0.08$ s, n=16, in KI) confirms this conclusion, and shows a similar slowing of K⁺ clearance in FHM2 KI mice in the absence (22 %) and presence (24%) of glutamate receptor blockers. Note that, actually, in both genotypes, the times constants of decay of I_K are similar in the absence and presence of glutamate receptors blockers (2.36 ± 0.10 s vs 2.38 ± 0.09 s in WT and 2.87 ± 0.10 s vs 2.95 ± 0.08 s in KI).

We have mentioned these findings in the Results section of the revised manuscript (p. 12, lines 269-274).

Minor points

Page 6. "[...] in layer 1 at 200 μm from" should be "[...] in layer 1, 200 μm from [...]"

Done. Thank you.

Page 7. "These data indicate that glutamate released by an action potential (AP) at cortical synapses is taken up more slowly by astrocytes in FHM2 KI compared to WT mice." There is no indication that the stimulus intensity used by the authors only evoked one action potential. This statement should be removed.

Line 167. "a train of 10 APs". This is not really what the authors use: they use a train of 10 stimuli but do not know how many action potentials the train evokes.

Thank you for pointing out this mistake, that has been corrected in the revised manuscript.

The authors should express t_{10}/t_1 using the same units for WT and KI. Currently they are 1.18 and 40%.

Probably the writing of the original manuscript was unclear and misleading because 1.18 referred to τ_{10}/τ_1 (50 Hz) for WT (the corresponding value for KI being 1.26) while 40% referred to the % increase of $\tau_{10(100\text{ Hz})}$ in FHM2 KI relative to WT. In the revised manuscript we have rewritten this section of the Results and hope that we have succeeded in describing our findings more clearly (p 8, lines 170-184).

The authors should cite the reference that provides a full description of the STC analysis (Scimemi and Diamond 2013, JoVe).

We have cited this reference in the revised manuscript in both Results and Methods sections. Thank you for the suggestion.

Line 206: "ir" should be "immuno-reactivity".

Done. Thank you.

Page 10: "the density of gold particles coding for GLT-1a". Please rephrase: the gold particles do not code for GLT-1a.

We have rephrased with "the density of GLT-1a gold particles".

Lines 241-242. "This correlates with, and likely reflects, the absence of NKA in cortical axon terminals (Cholet et al., 2002)." The expression of GLT-1a in axon terminals can also

happen in the absence of $\alpha 2$ NKA. Please rephrase.

We have rephrased with “ The similar density of GLT-1a in FHM2 and WT axon terminals correlates with, and likely reflects, the absence of $\alpha 2$ NKA in cortical axon terminals.” (p 10, lines 232-234)

Lines 236-238: "The reduction of GLT-1a density in the membrane of PAPs mirrors the about 50% reduction of $\alpha 2$ NKA protein level in cortical crude synaptic membranes from W887R/+ KI mice revealed by Western blotting (Leo et al., 2011) (data not shown)." Please rephrase.

We have rephrased with “The reduction of GLT-1a density in the membrane of PAPs is quantitatively similar to the reduction of $\alpha 2$ NKA protein level in cortical crude synaptic membranes revealed by Western blotting (about 50%)” (p 10, lines 227-230)

"For complete rescue one should have measured a 63% increase in CSD threshold in ceftriaxone-treated FHM2 KI mice, assuming that the facilitation of CSD induction in acute cortical slices from P30-33 saline-injected FHM2 KI mice is similar to that shown in slices from P34-35 KI mice (as suggested by the similar CSD threshold: 145 vs 149 ms)." Unclear; please rephrase.

Assuming that in acute slices from saline-injected P30-33 FHM2 KI mice (the age at which we measured the effect of ceftriaxone on CSD) the facilitation of CSD induction in KI relative to WT mice was similar to that previously measured in P34-35 KI mice (cf Supplementary Fig 2A of the revised manuscript showing a 39% reduction of CSD threshold in KI compared to WT, which corresponds to a 63% higher CSD threshold in WT relative to KI), then for complete rescue one should have measured a 63% increase in CSD threshold in ceftriaxone-treated compared to saline-treated FHM2 KI mice. If this calculation is correct, the 13 % increase in CSD threshold measured in ceftriaxone-treated mice corresponds to a $(13/63) \approx 21\%$ rescue of the facilitation of CSD induction.

Given that the important message to convey is that the rescue by ceftriaxone is relatively small (rather than the calculation of the fraction of rescue) and given also the space limits imposed by the maximal characters count, we have deleted from the manuscript the unclear sentence mentioned by the reviewer.

Figure 6A: the y-axis label is "volume", but it is unclear whether this is really the case. I thought this would be the band intensity value in the Western blot experiments. Please check

The referee is absolutely correct, and we apologize for the mistake. In the revised version, we have added new Western blots data (see replies to comment n. 3 and 9 of referee 2); and now all band intensities (for GLT1a, $\alpha 2$ Na⁺,K⁺ ATPase, xCT and, Kir4.1) were normalized to those of internal standards and expressed as % control.

I really would like to thank you for your patience and continued understanding during the revision process of your manuscript at EMBO Molecular Medicine. We have now at last received the enclosed reports from the referees that were asked to re-assess it. As you will see the reviewers are now globally supportive and I am pleased to inform you that we will be able to accept your manuscript pending the following final amendments:

In order to reply to referee 3's comments, please thoroughly discuss the limitations of your interpretation of the transporter currents as well as the limitations of the performed experiments. In addition, we would like you to tune down your claims in the abstract, introduction and results section to indicate more clearly that analysing STCs is not per se there same as analysing glutamate clearance, but it is indicative of.

I also wanted to congratulate you on providing a paragraph in the materials section entitled "Image processing for final illustrations". As EMBO Press stands for transparency in the publishing process, we are very happy about your initiative!

I look forward to seeing a revised form of your manuscript as soon as possible, ideally within 2 weeks.

***** Reviewer's comments *****

Referee #1 (Remarks):

The authors have addressed all comments raised by me in an adequate way in their revision. Therefore I recommend to accept the revised version for publication.

Referee #2 (Remarks):

The authors have been very thoughtful and thorough in their response to all the previous critiques.

Referee #3 (Comments on Novelty/Model System):

The authors analyze an interesting topic, related to the effect on astrocytes glutamate clearance and potassium homeostasis in mice affected by familial hemiplegic migraine type 2. My enthusiasm for this work is hampered by some fundamental inaccuracies in the interpretation of the transporter currents, which were correctly raised by Rev 1 and that do need to be addressed.

Referee #3 (Remarks):

In general, the authors have made significant efforts to address the concerns raised by all reviewers. Most of the points that I raised in my first review were addressed.

Despite this commendable effort, I noticed a fundamental problem with the interpretation of the transporter currents which was correctly pointed out by Reviewer 1. As a result, I think the manuscript still contains inaccuracies related to the interpretations of the transporter currents. The fundamental problem is that changes in the time course of the STCs **CAN** be indicative of changes in glutamate clearance. This does not mean that every time there is a change in the time course of the STCs this **MUST** be due to changes in glutamate clearance. For example, degrading synchronous release of glutamate will prolong the time course of the STCs without implying that these changes are due to an effect on clearance. The manuscript does not contain an analysis of glutamate clearance: it contains an analysis of STCs. It is therefore incorrect to conclude (lines 156-158) that " the rate of clearance of synaptically released Glu by cortical astrocytes is slower in

FHM2 KI compared to WT mice". The authors are aware of the literature on glutamate transporters from the Jahr, Diamond and Bergles labs, but their interpretation of this literature is inaccurate.

The measure of the time course of the potassium current should be performed in the presence of TBOA: from the profile of the currents shown in Figure 5, it looks like the authors did not use TBOA. This is a problem because the measurements of the decay of the recorded currents may be confounded by the time course of the STCs (shown in the first figures of this manuscript).

2nd Revision - authors' response

27 May 2016

Response to referee 3

In general, the authors have made significant efforts to address the concerns raised by all reviewers. Most of the points that I raised in my first review were addressed.

*Despite this commendable effort, I noticed a fundamental problem with the interpretation of the transporter currents which was correctly pointed out by Reviewer 1. As a result, I think the manuscript still contains inaccuracies related to the interpretations of the transporter currents. The fundamental problem is that changes in the time course of the STCs ***CAN*** be indicative of changes in glutamate clearance. This does not mean that every time there is a change in the time course of the STCs this ***MUST*** be due to changes in glutamate clearance. For example, degrading synchronous release of glutamate will prolong the time course of the STCs without implying that these changes are due to an effect on clearance. The manuscript does not contain an analysis of glutamate clearance: it contains an analysis of STCs. It is therefore incorrect to conclude (lines 156-158) that "the rate of clearance of synaptically released Glu by cortical astrocytes is slower in FHM2 KI compared to WT mice". The authors are aware of the literature on glutamate transporters from the Jahr, Diamond and Bergles labs, but their interpretation of this literature is inaccurate.*

We are fully aware that the time course of the STC does not exclusively reflect the time course of glutamate clearance and that, as stated by Diamond (2005), the "STC can be considered to represent the convolution of the time course of glutamate clearance with a filter representing the remaining elements of the system (axon activation and propagation, release asynchrony and electrotonic filtering)". Therefore, we agree with the reviewer that, **in principle**, changes in the time course of the STCs can be indicative of changes in glutamate clearance but do not necessarily reflect changes in glutamate clearance since they might reflect changes in the filter.

However, we maintain that it is correct to conclude, **on the basis of our findings**, that the rate of clearance of synaptically released glutamate by cortical astrocytes is slower in FHM2 KI compared to WT mice for the following reasons.

Assuming (ab absurdo) that this conclusion is incorrect, i.e. that the slower time course of the STC in FHM2 KI compared to WT mice does not reflect a slower rate of clearance of synaptically released glutamate by cortical astrocytes in FHM2 mice, then this implies that

i) the 50% reduction of glutamate transporters in the perisynaptic astrocytic processes of FHM2 KI mice (Figure 4) does not affect the time course of the STC; but this appears unlikely and contradicted by the fact that small (largely subsaturating) concentrations of the glutamate transporter inhibitor TBOA do slow down the decay of the STC in WT mice (Supplementary Fig 1A and Figure 9);

ii) the slower time course of the STC in FHM2 KI mice is entirely due to an increase in asynchronous glutamate release by nerve terminals and/or a slower axon propagation and/or a larger electrotonic filtering; but also this appears unlikely since it seems unlikely that a loss-of-function mutation in a Na,K ATPase that is almost exclusively expressed in astrocytes and is not expressed in cortical axon terminals (Cholet et al, 2002) causes increased asynchronous glutamate release by nerve terminals and/or a slower axon propagation. Moreover, previous studies analyzing the factors that shape the time course of the STC in hippocampal slices have shown that axon propagation, release asynchrony and glutamate diffusion contribute insignificantly to the time course of the STC and have suggested that astrocyte electrotonic filtering is the main mechanism that contributes to the STC time course besides glutamate clearance (Bergles and Jahr, 1997; Diamond, 2005). It seems unlikely that a 50% reduction of Na, K ATPase expression would affect the astrocyte electrotonic

filtering and quite unlikely that a larger astrocyte electrotonic filtering can account for the slower STC time course in FHM2 KI compared to WT mice, given that we measured similar passive properties of cortical astrocytes in FHM2 KI and WT mice. Moreover, the finding that the slowing of the STC in FHM2 KI mice was larger after a train of pulses compared to that after a single pulse (Fig 2) cannot be explained by a larger electrotonic filtering by astrocytes in KI mice. For these reasons we think that, as a whole, our data strongly support the conclusion that the rate of glutamate clearance is reduced in FHM2 KI compared to WT mice.

However, we have revised the Results section to make clear that the time course of the STC does not exclusively reflect the time course of glutamate clearance (p 5-6, lines 116-123). Moreover, we have considered possible alternative interpretations for the slower decay of the STC measured in FHM2 KI compared to WT mice and discussed why we consider these interpretations unlikely (p. 7-8, lines 161-174).

The measure of the time course of the potassium current should be performed in the presence of TBOA: from the profile of the currents shown in Figure 5, it looks like the authors did not use TBOA. This is a problem because the measurements of the decay of the recorded currents may be confounded by the time course of the STCs (shown in the first figures of this manuscript).

The decay of the STC is more than 2 orders of magnitude faster than that of I_K (cf time constants of 7.6-9.8 **milliseconds** for STC WT-KI vs time constants of 2.4-2.9 **seconds** for I_K WT-KI after a train of pulses at 50 Hz) and therefore the measurements of the decay of I_K are not confounded by the time course of the STC.

Corresponding Author Name: Daniela Pietrobon

Manuscript Number: EMM-2015-05944